# Dose-efficient cryo-electron microscopy for thick samples using tilt-corrected scanning transmission electron microscopy

Yue Yu [1,2] ✉, Katherine A. Spoth [1,3], Michael Colletta[1], Kayla X. Nguyen[1,4], Steven E. Zeltmann [1,5], Xiyue S. Zhang[1], Mohammadreza Paraan [2], Mykhailo Kopylov[6], Charlie Dubbeldam[6], Daniel Serwas[2], Hannah Siems[2], David A. Muller [1,7] ✉ & Lena F. Kourkoutis[1,7,8]

Cryogenic electron microscopy is a powerful tool in structural biology. In thick specimens, challenges arise as an exponentially larger fraction of the transmitted electrons lose energy from inelastic scattering and can no longer be properly focused as a result of chromatic aberrations in the post-specimen optics. Rather than filtering out the inelastic scattering at the price of reducing potential signal, as is done in energy-filtered transmission electron microscopy, we show how a dose-efficient and unfiltered image can be rapidly obtained using tilt-corrected bright-field scanning transmission electron microscopy data collected on a pixelated detector. Enhanced contrast and a 3–5× improvement in dose efficiency are observed for two-dimensional images of intact bacterial cells and large organelles using tilt-corrected bright-field scanning transmission electron microscopy compared to energy-filtered transmission electron microscopy for thicknesses beyond 500 nm. As a proof of concept for the technique's performance in structural determination, we present a single-particle analysis map at sub-nanometer resolution for a highly symmetric virus-like particle determined from 789 particles.

Cryogenic electron microscopy (cryo-EM) provides powerful insights into the study of biological systems by revealing molecular structures in their close-to-native environments[1–3]. Single-particle analysis (SPA) has enabled structural determination of purified macromolecular complexes up to atomic resolution[4,5]. Cryogenic electron tomography (cryo-ET) with subtomogram averaging has been developed to resolve macromolecular structures in biological contexts including within slices of whole cells[6,7]. Compared to SPA, fewer structures have been resolved at high resolution by cryo-ET with subtomogram averaging

with one of the main limitations being the increased specimen thickness for cellular structures compared to the preparations for the purified molecules. This increased sample thickness leads to an exponential decrease in the elastically scattered signal, especially at high sample tilts[8] or lower beam voltages[9]. In the conventional transmission electron microscopy (TEM) geometry, the imaging optics are placed after the sample, and chromatic blur in the post-specimen optics leads to a strong defocusing of the inelastically scattered electrons. Energy-filtered transmission electron microscopy (EFTEM) removes this blur caused

[1]School of Applied and Engineering Physics, Cornell University, Ithaca, NY, USA. [2]Chan Zuckerberg Institute for Advanced Biological Imaging, Redwood City, CA, USA. [3]Hauptman-Woodward Medical Research Institute, Buffalo, NY, USA. [4]Department of Physics, University of Oregon, Eugene, OR, USA. [5]PARADIM, Materials Science & Engineering Department, Cornell University, Ithaca, NY, USA. [6]New York Structural Biology Center, New York, NY, USA. [7]Kavli Institute at Cornell for Nanoscale Science, Cornell University, Ithaca, NY, USA. [8]Deceased: Lena F. Kourkoutis. ✉e-mail: yue.yu@czii.org; david.a.muller@cornell.edu

by inelastic scattering by simply removing the energy loss electrons from the image, but in doing so reduces the collected signal and dose efficiency compared to an ideal microscope[10,11]. Chromatic aberration correction could in principle correct some of this inelastic blur over a limited energy range and it is an ongoing topic of active research to improve the energy range, stability and resolution[12,13].

It has also long been recognized in the scanning transmission electron microscopy (STEM) geometry, where the electron beam is focused to a small spot and then rastered across the specimen, that post-specimen chromatic aberrations should not compromise the probe size. This is because in STEM the probe-forming optics are placed before the sample, and before any inelastic scattering can occur; thus, STEM imaging should be less susceptible to specimen-induced chromatic blurring (instead the chromatic blur in the post-specimen optics degrades the angular coherence of the diffraction pattern). Consequently, the possibility of studying micrometer-thick biological samples with STEM tomography has been explored both experimentally and theoretically, utilizing coherent signals, incoherent signals, and a combination of both[14–18].

Recent advances in the design of STEM detectors[19–22] have enabled rapid four-dimensional STEM (4D-STEM) data acquisition, where almost all of the scattered electrons are collected as two-dimensional (2D) images of convergent beam electron diffraction (CBED) patterns, and recorded over a 2D grid of probe positions, as sketched in Fig. 1a. 4D-STEM has simplified the implementation of other STEM phase imaging techniques such as integrated differential phase contrast (iDPC)-STEM[23] and has enabled sophisticated novel methods such as electron ptychography[24–26]. Efforts have been made to optimize these techniques for applications in structural biology studies. iDPC-STEM has generated the first SPA map of macromolecules embedded in vitrified ice by a STEM technique at near-atomic resolution[23]. Initial attempts at low-dose ptychography have been performed on purified virus-like-particles (VLPs) at nanometer resolution with a limited number of particles[24,25]. More recent demonstrations have shown that SPA of thin sections with ptychography can resolve protein structures at a sub-nanometer level[26], including a 5.8-Å SPA map of apoferritin reconstructed from ~11,500 particles. This performance is still worse than EFTEM and TEM, and suggests the resolution limit is not related to the instrument optics, but likely related to uncorrected beam-sample interactions. To date, both the iDPC and ptychography studies have focused on relatively thin samples that were optimized for SPA applications.

Here we describe how, in STEM geometry, a new dose-efficient phase-contrast imaging technique—tilt-corrected bright-field scanning transmission electron microscopy (tcBF-STEM)—could prove useful for imaging thick samples, while still providing comparable spatial resolution for thin samples. With this technique, we were able to resolve features in thick samples (roughly 500–800-nm thick) that were not visually discernible with EFTEM under comparable conditions in intact bacterial cells and large organelles. A 3–5× improvement in dose efficiency is observed with tcBF compared to EFTEM. Additionally, with a single-particle approach, we present a sub-nanometer-resolution three-dimensional (3D) map for a highly symmetric VLP from 789 particles as proof of feasibility for biomolecular structural determination with tcBF-STEM. Our earlier work on tcBF can be found in a series of short conference abstracts[27–31], but a detailed write-up of the method had been delayed by the illness and untimely passing of our colleague Lena F. Kourkoutis, and here we provide a more in-depth description. This technique is computationally much faster than iterative ptychography, so it could be used for live monitoring while collecting 4D-STEM data. We note this technique is starting to find applications in the development of low-dose ptychography for cryo-EM applications and materials science studies[26,32].

## Results

The starting point for tcBF-STEM is the collection of a 4D-STEM dataset (Fig. 1a) with a defocused probe, similar to what might be recorded for an out-of-focus ptychographic reconstruction[33]. This dataset records a detailed momentum distribution of the scattering at every recorded probe position. From this, almost all conventional STEM imaging can be reconstructed, while the scattering distribution can also be processed in more complex fashions than would be possible with traditional detectors to create new imaging modes.

### Producing a phase-contrast STEM image

Our goal is to produce a dose-efficient, phase-contrast image that can be easily integrated into the current cryo-EM workflow. For instance, a conventional bright-field (BF) TEM image could be reproduced by forming a STEM image from a small detector (for example, a single detector pixel) on the optic axis (Fig. 1b). This follows from the theorem of reciprocity[34], which states that "the electron intensities and ray paths in the microscope (including a specimen) remain the same if their direction is reversed and the source and detector are interchanged (that is, the electrons trajectories and elastic scattering processes have time reversal symmetry),"[35] as shown in the ray diagram of Fig. 1b. (Chapter 2.4 of ref. 35 provides a more detailed discussion.) This reversal also means the condenser aperture in a TEM column becomes the objective aperture when operated in STEM mode, and the pre-field rather than the post-field of the objective lens is used.

Producing a phase-contrast image by traditional STEM approaches is very dose inefficient. In the above example, the angular range of the detector must be kept small to preserve a coherent signal—the collection angle of the STEM detector is equivalent by reciprocity to the illumination angle in TEM. However, in STEM mode, a single small detector collects only a tiny fraction of the incident beam. For instance, a 0.1-mrad-wide collector with a 10-mrad probe convergence angle achieves just 1 in 10,000 collection efficiency. In contrast, a TEM with a 0.1-mrad illumination convergence and a 10-mrad post-specimen objective aperture has near-perfect collection efficiency in a thin sample.

To improve the collection efficiency in STEM, we could increase the collection angle of the detector, but this will eliminate the phase-contrast signal (a much weaker amplitude contrast in an incoherent image will still be present—for example, see chapter 3 of ref. 35). This is because the phase-contrast signal is only measurable when there is a phase shift on the lens, but the phase shift from aberrations (including defocus) also generates an image shift that is different for each angle. In other words, simply summing over a wide range of angles leads to a blurred image.

tcBF-STEM seeks to undo this aberration-induced image shift and thus allows us to utilize the full range of angles within the BF disk. For tcBF-STEM, each pixel within the BF disk functions as a coherent BF detector if it subtends a sufficiently small collection angle. From Fig. 1b, we see that reciprocity implies that STEM images produced by off-axis detector pixels are each equivalent to BF-TEM images formed with tilted illumination. (There are some important differences for inelastic scattering[36] that we discuss below, and more details are provided in Methods, where we follow the image analysis framework laid out by Rose[36].) These equivalent beam tilts in TEM or detector shifts in STEM give rise to image shifts that depend on the aberration function[37,38].

If the dominant aberrations are defocus and coma, the STEM images produced by the off-axis detector pixels have similar contrast transfer functions (CTFs) to the on-axis pixel[35] (except toward the edge of the aperture; Methods), and so after removing the image shifts they may be summed to produce a single image that retains coherent phase contrast while using almost all of the incident beam, that is, maintaining a similar dose efficiency to TEM. This process of measuring and correcting the shifts between images formed from the individual detector pixels in a 4D-STEM dataset is the basis of tcBF-STEM.

Such an image shift is demonstrated on two images recorded with two off-axis detector pixels (Fig. 1c,d) on a gold-shadowed carbon test specimen. The shifts are measured and corrected on a (detector)

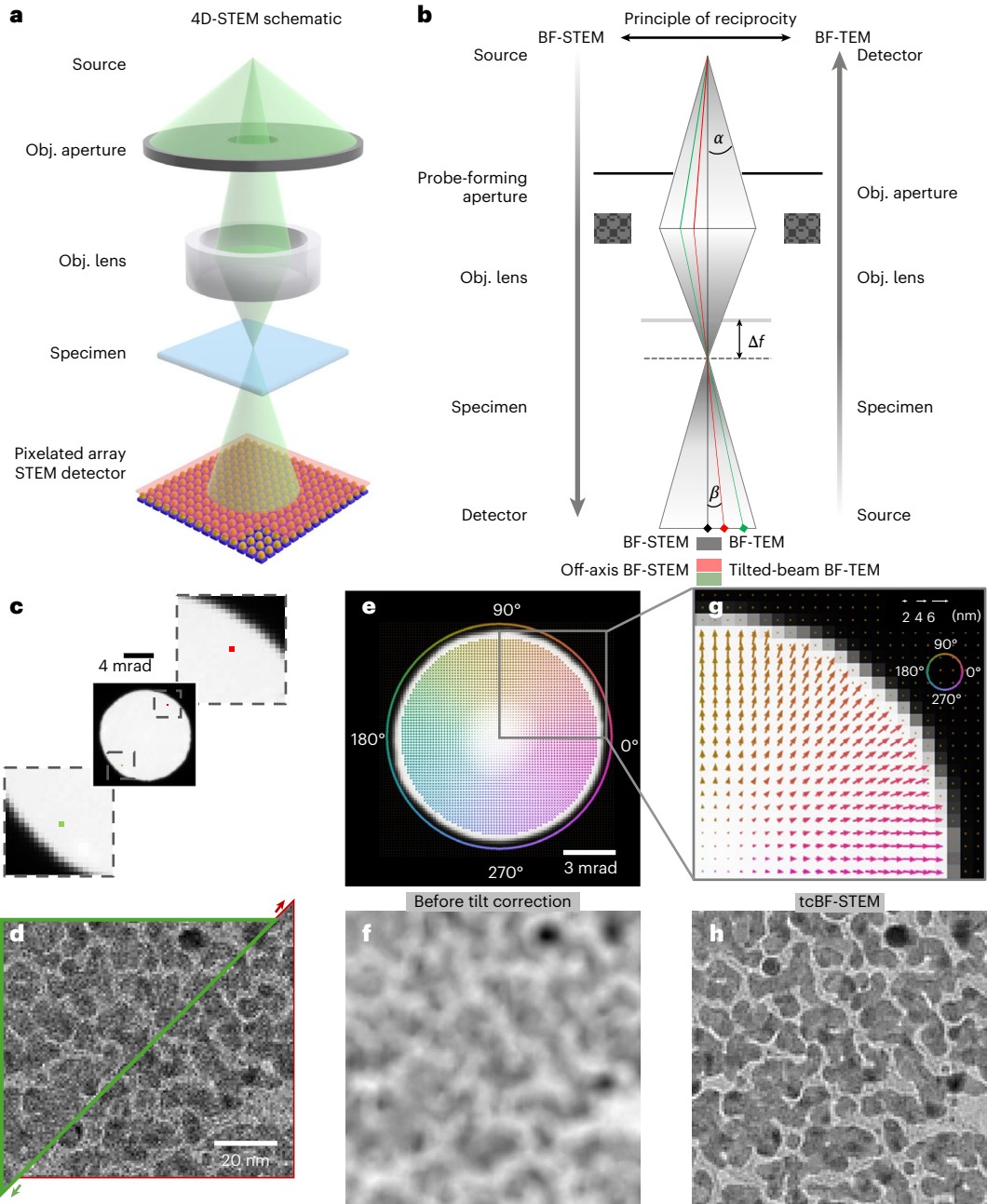

**Fig. 1 | Direct phase contrast with tcBF-STEM. a,b**, Direct phase-contrast imaging with 4D-STEM: tcBF-STEM uses a pixelated STEM detector to collect the entire CBED pattern (**a**). Each detector pixel within the BF disk is a coherent BF-STEM detector, although they are located off the optical axis. By reciprocity (**b**), off-axis BF-STEM (top down) is equivalent to tilted-illumination TEM (bottom up). Similarly to BF-TEM, defocus ($\Delta f$) is applied to introduce phase contrast. The convergence angle, α, in STEM is equivalent to the collection angle in TEM, and the STEM collection angle, $\beta$, on a single pixel in tcBF would be equivalent to the illumination angle in TEM. **c,d**, For a standard gold-on-carbon sample and a defocused probe, integrating the signals collected by two off-axis detector pixels (red and green) in **c** produces two images with relative shifts between them (**d**).

**e–g**, The relative shifts are observed and measured across all 256 × 256 individual detector pixel images. For every detector pixel, the image shifts determined through cross-correlation with the on-axis detector pixel are shown by the arrows overlaid on the averaged CBED pattern in **e** with a zoomed-in and binned view in **g**. The arrows are color coded corresponding to the shift directions. Integrating the full forward-scattered BF signals without correcting for the angle-dependent shifts results in blurring (**f**) due to the defocus. **h**, A tcBF-STEM image is generated by summing the images after shift correction. In a tcBF-STEM image, the SNR is increased compared to **d** and the blurring due to defocus (**f**) is corrected. This before-and-after correction behavior (**f**–**h**) was consistently observed across all *n* = 24 tcBF images reported in this paper.

pixel-by-pixel basis. Figure 1e,g illustrates the resolved shift map overlaid on the averaged CBED pattern. Each individual image, after shift correcting, is then combined to create the final tcBF-STEM image (Fig. 1h). Compared to the BF images formed by a single detector pixel (Fig. 1d), the tcBF-STEM image has a notably improved signal-to-noise ratio (SNR) because almost all the signal-relevant channels are utilized.

Furthermore, compared to the image formed by directly integrating over the full BF disk (Fig. 1f), tcBF preserves a sharper image. When reconstructing a tcBF-STEM image, a simultaneous measurement of the probe aberration function can be obtained. In fact, one of the early applications of a shift analysis of 4D-STEM datasets was for aberration measurement[39], by analogy with the TEM beam tilt methods[37].

In tcBF-STEM, like in conventional BF-TEM, defocus is deliberately introduced to enhance contrast. Consequently, in Fig. 1e,g, the shift magnitudes are linearly proportional to the defocus and the off-axis angles, and are oriented outwards. The linearity of the shift with angle also makes it possible to measure the depth of objects by the resulting parallax effect[22].

The presence of the aperture complicates the analysis compared to aperture-free TEM, but the end result for tcBF is a similar-looking CTF that has an information limit at double the aperture size (Methods). This is the same information limit cutoff for iDPC and BF ptychography, although the shapes of the CTF are very different, and double the information limit of BF-STEM. As we will discuss, iDPC is less efficient than tcBF at transferring low-frequency information, although it is simpler to interpret.

Moreover, tcBF-STEM has an advantage over EFTEM for thick samples. The post-specimen lenses for EFTEM are the image formation lenses so chromatic aberrations in the post-specimen lenses degrade the image resolution. However, for tcBF-STEM, the post-specimen lenses simply transfer an image of the diffraction pattern so chromatic aberrations result in a small loss of angular resolution—that is, a small increase in the effective detector pixel size and hence a reduction in coherence. In a thick sample, most electrons undergo both elastic and inelastic scattering, but the elastic contrast is preserved when scattered to the inelastic channels[36,40]. This is largely because the most-likely inelastic scattering events are very delocalized compared to the elastic scattering, leading to weak and low-frequency modulations of the real-space signal, and only a small blurring of the angular distributions, with 0.07 mrad for the most probable inelastic scattering angle to 0.7 mrad for the median inelastic scattering angle (see sections 3.3.1.5 and 5.5.3 of ref. 8).

In general, to preserve coherent contrast, the collected angular range should be small compared to the convergence angle. How small is a function of spatial frequency and requires detailed simulations, but for low spatial frequencies and a typical convergence angle of 4–8 mrad, we would expect a 1–2-mrad angular blur could be tolerated (Extended Data Fig. 1). This would correspond to about three independent inelastic events, which is roughly what would be expected in a micron-thick ice-embedded sample given an inelastic mean free path (MFP) of ~310 nm (ref. 41). Multiple elastic scattering with an ~10-mrad characteristic scattering angle per event places another expected limit. With an elastic MFP of ~800 nm (Methods), only ~30% of the usable coherent signal would remain at a micron thickness.

## Fast data acquisition with upsampling and the CTF of tcBF-STEM

In tcBF-STEM the number of pixels in the reconstructed image can be made much larger than the number of diffraction patterns recorded, as at a finite defocus each diffraction pattern contains information about an extended region of the sample. This trade-off between real space and reciprocal-space sampling helps speed up the data collection because the multi-pixel detectors used for tcBF tend to be slower to read out than single-pixel sensors. For instance, if each diffraction pattern contains information from an 8 × 8-pixel patch in the upsampled reconstruction, the data collection rate is sped up 64-fold, so a 10-kHz detector frame rate becomes a 640-kHz image pixel rate. The recovery of information beyond the limits of the real-space probe sampling uses the information collected in the shadow images in the diffraction plane. The information retrieval is achieved by measuring the image shifts to sub-scan-pixel precision, followed by upsampling of the images before correction of the shifts. To understand the implementation of this upsampling technique, we start with a demonstration on a standard gold-on-carbon sample. This same approach is then also applied in the data reconstruction workflow for all the examples shown in the paper.

Figure 2 shows a tcBF-STEM dataset acquired with 256 × 256 scan positions, spaced 8 Å apart on a gold-shadowed carbon test specimen.

With the chosen defocus value and scan step size, we expect over 90% information overlap collected in the reciprocal space. We discuss the considerations that influence the choice of the defocus (and thus the probe size) and the scan step size in the 'Discussion' and Extended Data Fig. 2. This information surplus is used to upsample a tcBF image. Details on upsampling are available in Methods and Supplementary Figs. 1–3. Figure 2a,b demonstrates the process of upsampling. Figure 2d is the final upsampled result with sub-scan-pixel features resolved compared to the original image (Fig. 2c). In Fig. 2e, Thon rings and the 2.3-Å spacing of gold are recovered through upsampling. A fast Fourier transform (FFT) radial average profile (Fig. 2g) is shown to confirm that upsampling restores the information beyond the scan Nyquist frequency without altering the information within the frequency range. At an 8-Å scan spacing, the sampling corresponds to a Nyquist frequency of 16 Å, but upsampling enables information transfer down to the electron-optical information limit (measured at 2.3 Å and with a theoretical diffraction limit of 1.8 Å). This process extends the effective information transfer up to seven times the real-space Nyquist limit (16/2.3 ≈ 7), accelerating data acquisition by a factor of 49. The upsampling capability of tcBF-STEM effectively speeds up data collection for this method.

The number of pixels in the image is separate from the optical resolution limit. Notably, with the α = 5.5-mrad convergence semi-angle, the information transfer limit at a cutoff of 1α corresponds to 3.6 Å and 1.8 Å at 2α. The 2.3-Å spacing we observed exceeds the 1α cutoff but is just within the 2α limit. The calculated phase-contrast transfer function (PCTF) shown in Fig. 2f shows tcBF has a similar-looking PCTF to BF-TEM but with an information limit at double the aperture size, boosted by the information transfer from the off-axis detector pixels. Details are available in Methods and Extended Data Figs. 3–5, including a discussion of practical limits.

## Comparison of cryo-tcBF-STEM and EFTEM for imaging thick samples

To compare the performance of tcBF-STEM with EFTEM, the most widely adopted imaging technique in cryo-EM, we performed successive imaging under the same incident dose with the two techniques on various thick specimens, including intact bacterium cells and large cellular organelles. As the samples are much thicker than the depth of field, quantitative metrics on the full projection convey less information than they would for thin sections, or individual molecules at different depths (which can be determined by the parallax shift in tcBF). Instead, we present comparative cases with different acquisition orders, and different defocus choices for the two techniques.

For a qualitative comparison of EFTEM and tcBF-STEM in unstained, thick specimens, we imaged the same area in a mitochondrion from the HEK293T cells in succession with the two techniques (Fig. 3a,b). A full thickness map of this region can be found in Extended Data Fig. 6 and the thickness is estimated with the inelastic MFP using the EFTEM dataset. In the thinnest part of the sample at the organelle's edge (~500 nm), the membrane bilayers are similarly resolved with both methods. However, in thicker regions (~550 nm), tcBF-STEM still clearly shows the bilayers (Fig. 3c) of the mitochondrial inner membranes, whereas with EFTEM these features are less visible (Fig. 3d). In the thickest portion of the image (~600 nm), tcBF can still resolve some parts of the inner membranes (Fig. 3c), whereas in the EFTEM image (Fig. 3d) these features are hardly discernible. However, it is worth noting that warping of the sample during exposure might alter the orientation of the membrane relative to the beam, potentially altering its contrast. Another possibility is that the inherent range of tilts in the tcBF collection (up to ~7 mrad) enhances contrast for a broader range of membrane orientations. Even though no high-resolution information is compared, the image acquired first still introduces radiation damage and conformational changes before the second. In this case STEM imaging was acquired first, and multiple scenarios (Table 1, Extended

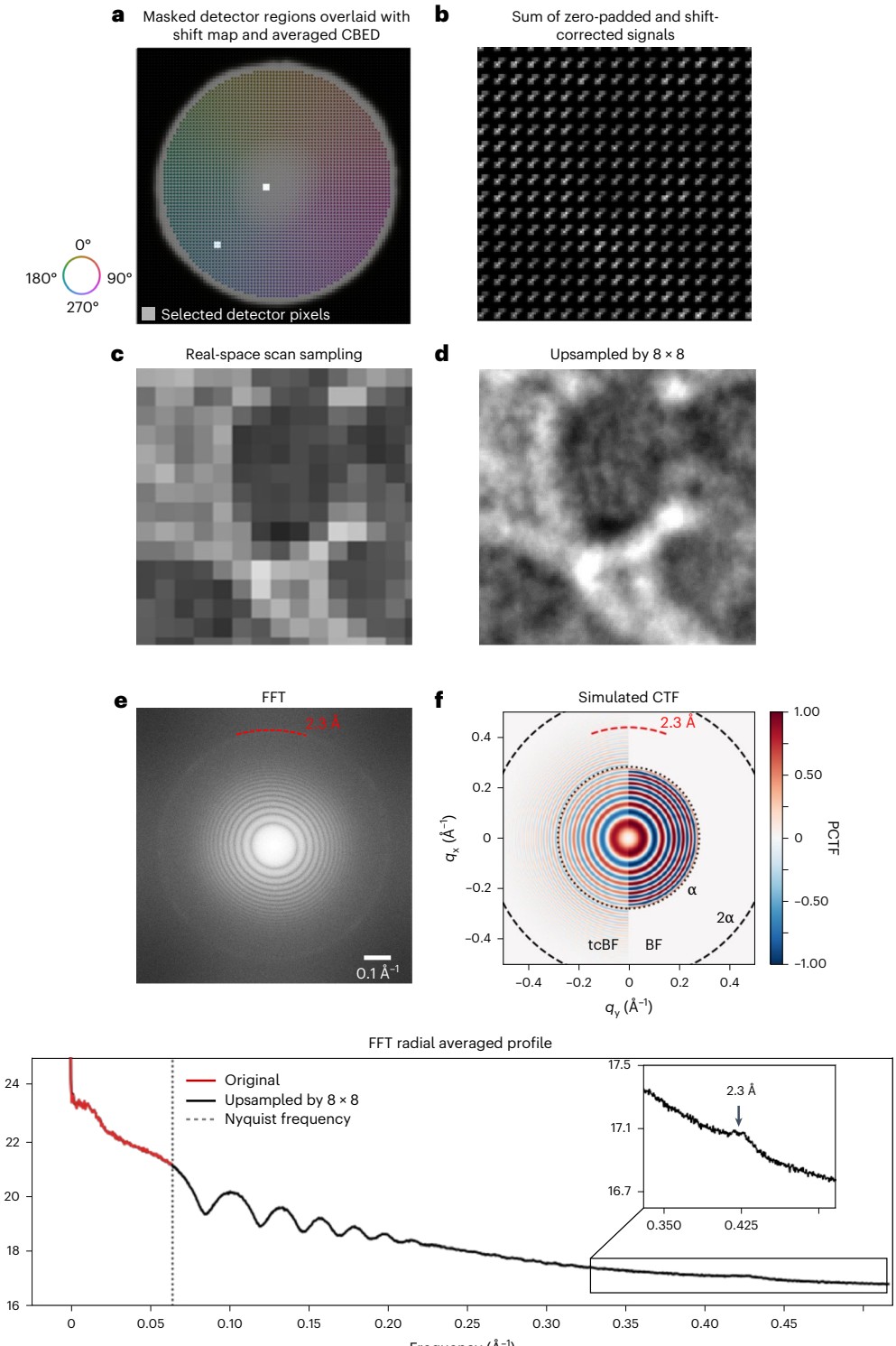

**Fig. 2 | Sub-scan-pixel shifts allow upsampling beyond scan Nyquist limit.**
Upsampling by 8 speeds up data acquisition by 64-fold. Upsampling for tcBF-STEM of a gold-on-carbon combined test sample is accomplished by exploiting the image shifts from different detector pixels as a result of defocus (and higher-order aberrations). **a**–**d**, The colored arrows show the shift measured for the scanned images synthesized at each detector pixel inside the BF disk. Scanned images formed by the two white pixels on the detector shown in **a** will be shifted from each other. Correcting for these shifts and accumulating signals collected from the selected detector regions fills in different regions of the scanned image (**b**) at a spacing finer than the recorded probe positions, demonstrating the first step of a complete upsampling. A tcBF-STEM image (**c**) is collected with a defocused probe at an 8-Å scan step size. In the upsampled tcBF-STEM image (**d**),

additional sub-scan-pixel features are resolved compared to the original image (**c**). All $n = 24$ tcBF images included in this paper are upsampled. **e**, Experimental power spectrum from the full image of the test sample showing Thon rings and the 2.3-Å ring of gold lattice spacing beyond the scan Nyquist frequency (1/16 Å$^{-1}$, black box) are recovered by upsampling. **f**, The calculated PCTF for a tcBF image after shift correction shows twice the information limit compared to the BF image formed using only the axial detector pixel, as a result of exploiting off-axis information. The simulation uses 5.5-mrad convergence semi-angle probe-forming aperture ($\alpha$), 300-kV acceleration voltage and 700-nm defocus. **g**, An FFT radial average profile shows that upsampling recovers information beyond the scan Nyquist frequency without altering the signal within the electron-optical information limit.

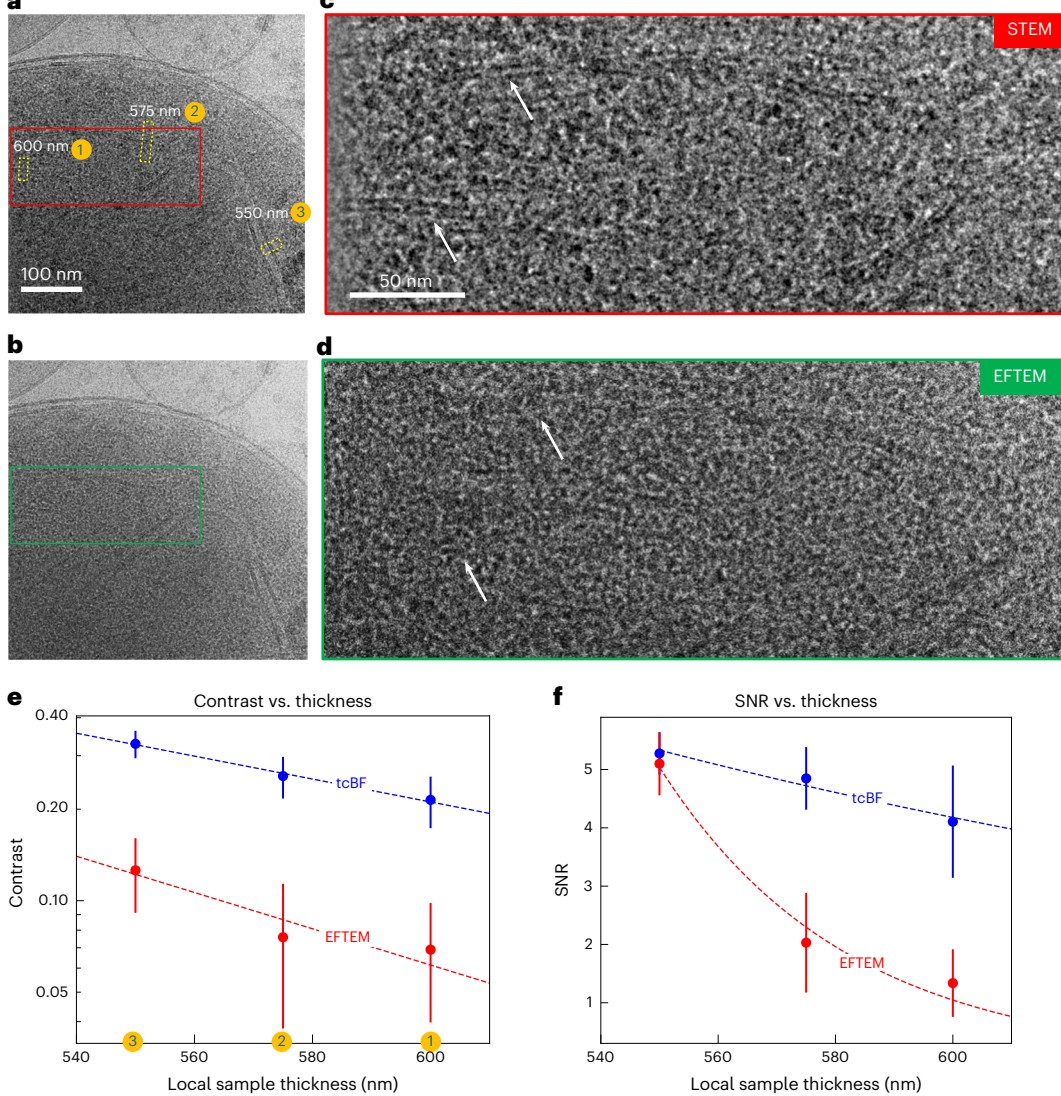

**Fig. 3 | Comparison of tcBF-STEM and EFTEM for imaging thick cryo biological samples. a–d**, Comparison of the performance of tcBF-STEM (**a**) with EFTEM (**b**) on thick samples. **a** and **b** are the images acquired from the same area in a mitochondrion, and **c** and **d** show a zoomed-in view. White arrows indicate bilayers. The dose measured over the vacuum is 14 e⁻/Å², and the acceleration voltage is 300 kV for both acquisitions with additional details in Table 1. For a more quantitative analysis, line profiles were taken across the membrane in the yellow-dashed rectangles marked on **a**, both for tcBF and EFTEM. A zoomed-in view of **a** and line profiles are shown in Supplementary Fig. 6 and Extended Data Fig. 8. **e**,**f**, From the fit to the line profiles, we extracted the fringe contrast with respect to the background, and the fringe signal to background noise, which are summarized in **e** and **f**, respectively, again showing better performance for tcBF over EFTEM with increasing sample thickness. The dashed lines in **e** and **f** are exponential fits. Sample thickness is estimated from 470 nm to 620 nm using the fraction of electrons remaining in the EFTEM image. The thickness map is given in Extended Data Fig. 6. Additional comparisons of tcBF versus EFTEM for a range of different samples and acquisition conditions are available in Table 1, Extended Data Fig. 7 and Supplementary Figs. 4 and 5. The error bars in **e** and **f** are the root-mean-squared errors between the measured line profiles and curve fits described in Extended Data Fig. 8. The number of data points in the EFTEM and STEM line profiles for each thickness in **e** and **f** are $N_{EFTEM}$ = 133, 289 and 116 and $N_{STEM}$ = 90, 194 and 53.

Data Fig. 7 and Supplementary Figs. 4 and 5) were compared where EFTEM images were acquired first.

As an attempt to quantify these visual observation trends for the two techniques, we compare the dose efficiency, the fringe contrast and the SNR of the membrane bilayer features. Under the same incident dose, and using direct electron detectors for both techniques, the collected electron counts were compared for dose efficiency. In the thickest parts of this sample, and at the same incident dose, the recorded electrons in the EFTEM image dropped to ~14% of the incident dose, but the tcBF still retained 50% of the incident dose, that is, almost 3.6 times more electrons remaining for tcBF. Similar trends in dose efficiency were also observed in multiple samples (Extended

Data Fig. 7 and Supplementary Figs. 4 and 5) and summarized later in Fig. 5a. Figure 3e shows the higher measured fringe contrast for tcBF compared to EFTEM from the three locations marked in Fig. 3a, with a larger view in Supplementary Fig. 6 and line profiles in Extended Data Fig. 8. Comparing the fringe intensity to the background fluctuations in Fig. 3f gives an SNR metric, which again shows a more favorable SNR for tcBF than EFTEM in the thicker regions.

We observed similar trends in thick specimen regions for tcBF, where features other than membranes were compared. Extended Data Fig. 7 shows EFTEM and tcBF images of an intact *Escherichia coli* cell. With tcBF, features within the cell's interior (Extended Data Fig. 7d) are effectively resolved, while in EFTEM the same features

**Table 1 | Summary of specimens, experimental parameters and a comparison of the dose efficiency of EFTEM and tcBF**

|  | Fig. 3a–d | Extended Data Fig. 7 | Supplementary Fig. 4 | Supplementary Fig. 5 |
|---|---|---|---|---|
| Specimen | Mitochondrion | E. coli | Vesicle | E. coli |
| Average thickness | 547 nm | 673 nm | 596 nm | - |
| Incident dose | 14 e⁻/Å² |  |  | 0.5 e⁻/Å² |
| Acquired first | tcBF | EFTEM | tcBF | STEM |
| EFTEM image pixel size | 2.37 Å |  |  | 12.96 Å (bin 4×4) |
| tcBF image pixel size | 3.59 Å (upsample 8) |  |  | 10.8 Å (upsample 4) |
| EFTEM defocus | 3,974.9 nm | 3,743.8 nm | 2,849.2 nm | 4,000 nm (nominal) |
| tcBF defocus | 1,943.6 nm | 4,252.9 nm | 2,825.0 nm | 4,000 nm (nominal) |
| Fraction of incident electrons in image - EFTEM | 0.171 | 0.114 | 0.143 | - |
| Fraction of incident electrons in image - tcBF | 0.533 | 0.403 | 0.522 | - |
| STEM semi-convergence angle α | 7 mrad |  |  | 2 mrad |
| STEM probe size | 1.6 Å |  |  | 6 Å |
| STEM iDPC depth of field | 80 nm |  |  | 984 nm |
| EFTEM detector | Falcon 4i detector |  |  | Gatan K2 |
| EFTEM energy filter | Selectris X |  |  | Gatan Tridium |
| tcBF/STEM detector | EMPAD |  |  |  |

For tcBF, the depth of field depends on spatial frequency within the aperture, not the aperture itself. The iDPC depth of field is calculated by $2\lambda/\alpha^2$; probe size is calculated from the diffraction limit $0.61\lambda/\alpha$. The EFTEM images were recorded with dose fraction and motion corrected with MotionCor2 (ref. 48) and defocus measured with CTFFIND4 (ref. 43). The STEM defocus is measured with tcBF. Thickness is measured from the EFTEM images, with and without energy filtering. The fraction of incident electrons is measured by comparing the incident and the recorded number of electrons with direct electron detectors for both methods.

(possibly condensates or surface contamination) are discernible but less prominent. In Supplementary Fig. 4a,b, images with tcBF and EFTEM of a vesicle were acquired, and again the features in the thick region (indicated by the arrows) are clearer with tcBF compared to EFTEM. For the comparisons in Supplementary Fig. 4, EMTEM imaging was performed before STEM. For another comparison on *E. coli* at a low dose of 0.5 e⁻/Å² (Supplementary Fig. 5), tcBF is capable of resolving features that are otherwise indiscernible with EFTEM. Table 1 provides a summary of the specimens, experimental parameters and dose efficiency comparisons.

### Single-particle analysis 3D reconstruction with cryo-tcBF-STEM imaging

To quantitatively assess the current performance of tcBF-STEM for molecular structure analysis, we performed SPA of bacteriophage PP7 coat protein, achieving sub-nanometer resolution using a generic cryoSPARC SPA workflow[42]. For this analysis, 789 particles were extracted from 19 tcBF-STEM micrographs. Figure 4a displays a cropped representative micrograph with 2D class averaging of the particles in the inset. As shown in Fig. 1f, the CTFs for tcBF and TEM share the same zero-crossing locations up to 1α information transfer. Therefore, we used CTFFIND4 (ref. 43), a tool widely used for conventional TEM, to estimate the CTF for each tcBF micrograph. No CTF refinement was performed due to the limited number of particles. Approximately 200 particles were manually picked to generate the template for template picking, and 789 particles were selected. The selected classes were then used for ab initio model generation, as a starting model for the homogeneous refinement with icosahedral symmetry applied. Figure 4b presents the cryo-EM density map sharpened with a Guinier B factor of 351 Å² based on the Guinier plot analysis of the 3D reconstruction. A zoomed-in view of the EM density with X-ray crystal structure of the particle docked inside (Protein Data Bank (PDB) 1DWN)[44] focused around the helices is shown in Fig. 4c. Fourier shell correlation (FSC) indicates a 7-Å resolution with a cutoff of 0.143 using the dynamic mask in cryoSPARC and 9.6 Å with no mask (Fig. 4d).

VLP PP7 possesses an icosahedral symmetry with triangulation number T = 3 and a theoretical molecular weight of 2 MDa, containing a high number of repeated units per particle, which allows efficient structural averaging with a smaller number of particles. The 7-Å resolution demonstrates the feasibility of using tcBF-STEM for structural analysis at a resolution that can resolve some secondary structures such as alpha helices. We also have preliminary experimental results suggesting a comparable resolution is achievable with the current ptychography algorithms with a similar number of particles. On the other hand, state-of-art EFTEM imaging under similar doses and the same accelerating voltage can achieve 3.3-Å nominal resolution with 900 particles (Extended Data Fig. 9a–c), albeit with more sophisticated data correction steps that have not yet been applied for STEM.

### Discussion

A key advantage of tcBF is in imaging thick specimens, as it is relatively insensitive to specimen-introduced energy losses. tcBF-STEM stands out as a STEM technique due to its high-dose efficiency, as it takes advantage of nearly all the forward-scattering electrons, making it a potentially powerful imaging technique for studying thick, dose-sensitive specimens, showing a twofold dose advantage over EFTEM at 400 nm growing to fivefold at 800 nm.

Figure 5a summarizes the measured fraction of electrons collected for tcBF and EFTEM from the different samples given in Table 1 (Fig. 3, Extended Data Fig. 7 and Supplementary Figs. 4 and 5), where tcBF collects a factor of 3–3.5× more signal, an advantage that grows with thickness. We expect both signals to decay exponentially with thickness ($t$), that is, $\exp(-t/\lambda_{in})$ for EFTEM and $\exp(-t/\lambda_{el})$ for tcBF. Fitting to the tcBF data in Table 1, we find the elastic MFP is $\lambda_{el} = 830 \pm 50$ nm assuming an inelastic MFP $\lambda_{in}$ of 310 nm (Fig. 5a), close to the expected $\lambda_{el} = 774 \pm 45$ nm (Methods). Some sense of the relative dose efficiency of the two approaches is given by the ratio of these two exponential decays, that is, $\exp(t/\lambda_{eff}) = \exp(-t/\lambda_{el})/\exp(-t/\lambda_{in})$ where $-1/\lambda_{eff} = 1/\lambda_{el} - 1/\lambda_{in}$, so $\lambda_{eff} \approx 500$ nm. This gives the factor of 3 advantage for tcBF at 550 nm, and it grows to 5× at ~800 nm. Beyond a

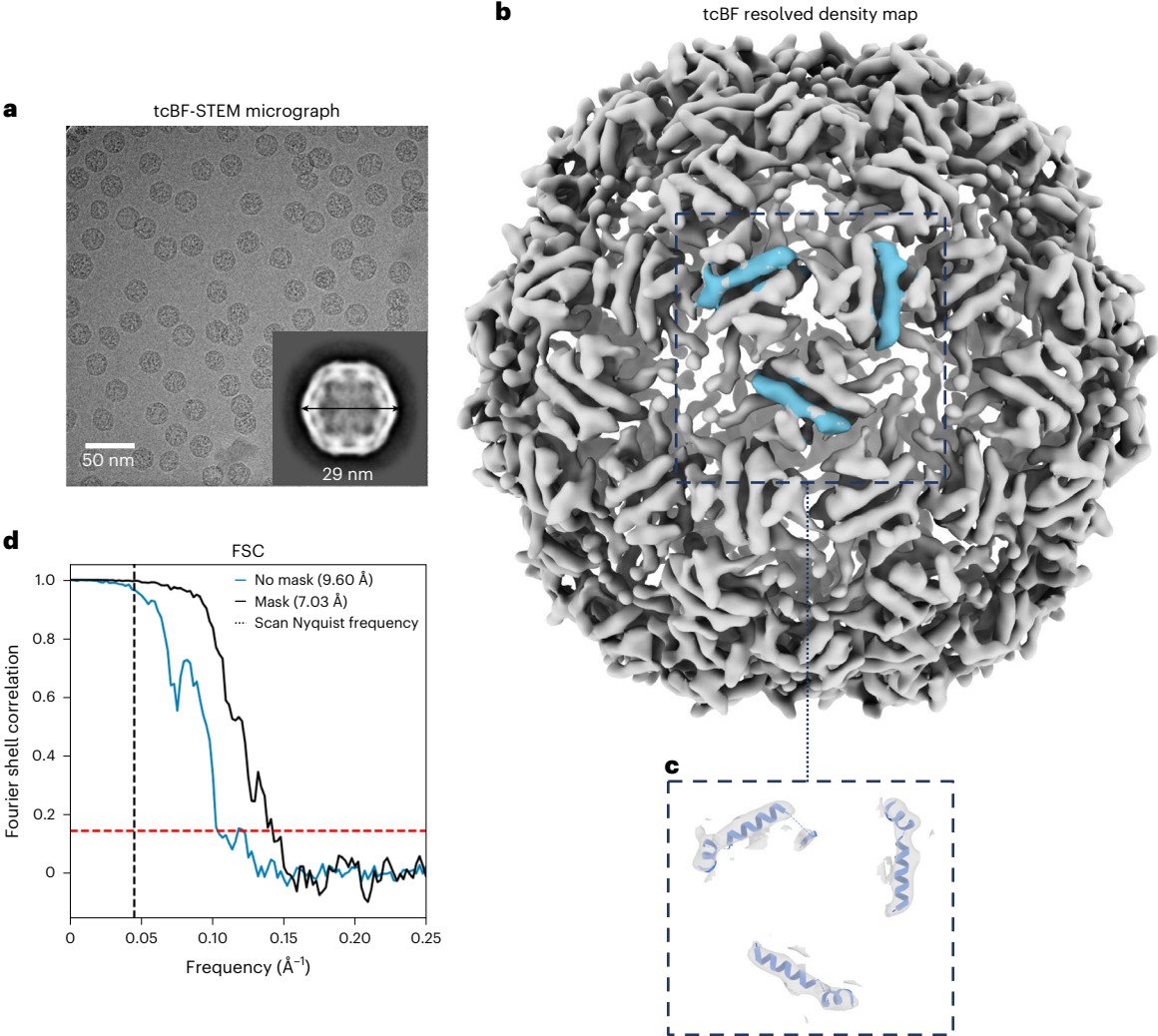

**Fig. 4 | 3D reconstruction of bacteriophage coat protein from tcBF-STEM single-particle analysis. a–c**, Single-particle analysis 3D reconstruction from tcBF-STEM imaging of hydrated vitrified coat protein of bacteriophage PP7. A representative upsampled tcBF-STEM image at 300 kV with 11-Å scan step size and a total dose of 45 e⁻/Å² is shown in **a** with a 2D class average in the inset. The 3D density map resolved from 789 particles is shown in **b** with a zoomed-in view of the PDB 1DWN model fit inside focused on helices in **c**. **d**, A 7-Å resolution is reached based on the FSC with a cutoff of 0.143.

thickness of one elastic MFP, much of the phase-contrast signal will be lost to multiple elastic scattering and leaving mostly amplitude contrast. This identifies an effective dose advantage window for tcBF over EFTEM for thicknesses beyond ~400 nm.

For low spatial frequencies, the dose efficiencies for tcBF and EFTEM (axial BF) can be compared directly because of their similar CTFs (Fig. 5b). STEM methods that also collect the entire BF disk such as DPC and iDPC can also be compared after accounting for their differences in information transfer as a function of spatial frequency. This is captured by the detective quantum efficiency (DQE) of the imaging system (equations (15)–(17)). Figure 5c shows that tcBF is more efficient at low spatial frequencies. The iDPC CTF and DQE peak at zero defocus and degrade with increasing defocus[45], unlike EFTEM and tcBF, which require a defocus or phase shift to transfer information.

Figure 5b compares the CTF for tcBF and DPC, which are the recorded signals we need for estimating the SNRs for the two methods. The axial BF CTF cuts off at $\alpha$, while the DPC and tcBF information limits extend to $2\alpha$. The damping envelope for tcBF follows the classic double-overlap form expected from summing over the tilted CTF functions in Extended Data Figs. 4 and 5. The DPC signal peaks close to $\alpha$ and is suppressed at low frequencies compared to the defocus-optimized tcBF but is more efficient from $\alpha$ to $2\alpha$. The iDPC CTF has the same shape

as the damping envelope but does not reflect the true information transfer. The iDPC CTF is obtained by dividing the measured DPC signal by spatial frequency, which also amplifies noise by the same proportion, resulting in a vanishingly small SNR at low spatial frequencies (see Methods for analytic derivations).

The DQE values for tcBF and iDPC are proportional to squares of the CTFs plotted in Fig. 5b (equation (16); for it is the DPC CTF and not the iDPC CTF that determines the iDPC DQE). While both approaches have the same information limit, tcBF has a higher information transfer at low spatial frequencies where much of the relevant structural information in a thick sample is located. In the language of ptychography[46], tcBF is able to access both the double-overlap and triple-overlap regions, while DPC and single-side-band (SSB) ptychography access only the double overlap (Methods). tcBF is also able to surpass the real-space scanning Nyquist limit, offering a possibility for more-rapid data acquisition by trading detector pixels for real-space positions, important for outrunning environmental noise in cryogenic experiments.

The CTF and DQE curves of Fig. 5 need to be modified for samples thicker than the depth of field as molecules at the top of the sample will have a different CTF from those at the bottom and would need to be corrected separately (Supplementary Fig. 7). If we consider the

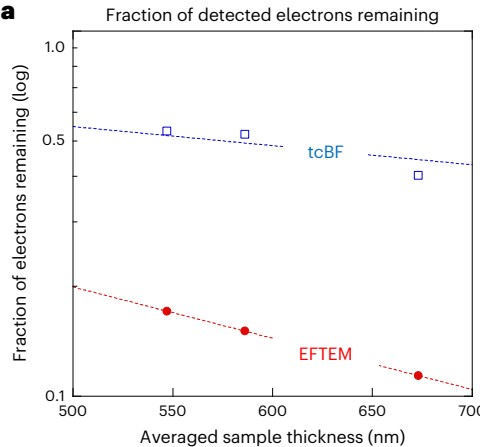

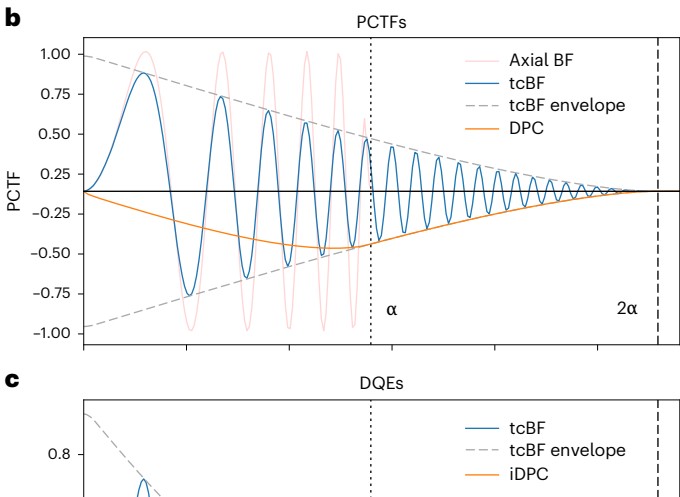

**Fig. 5 | Signal retention and contrast transfer efficiency of tcBF-STEM compared to (EF)TEM and DPC. a**, Collection efficiency of EFTEM (no objective aperture) and tcBF (7-mrad aperture), showing the measured fraction of electrons left in the image compared to the incident beam as a function of sample thickness from the datasets in Table 1. The sample thickness is determined from the EFTEM fraction, assuming an inelastic MFP of 310 nm. From this, the decay of the unfiltered tcBF images gives an elastic MFP of 830 ± 50 nm. **b**, Comparison of the CTFs for tcBF, axial BF and in-focus DPC for a 5.5-mrad probe-forming aperture, $\alpha$. The iDPC CTF has the same shape as the tcBF damping envelope. **c**, DQE for tcBF and iDPC. The DPC DQE is the same as that for iDPC.

sample to be uniform through the thickness, the effective CTF can be approximated by simply averaging though the projection. In this case the high frequencies are lost (Supplementary Fig. 8), roughly following the simple depth of field equation $\Delta z \approx \lambda/\alpha^2$, where $\alpha$ is the spatial frequency of interest rather than the probe-forming aperture cutoff. This means measuring the projected amorphous speckle pattern will give a worse estimate of resolution than measurements from individual molecules.

The phase-contrast CTF curves are only the first term in a Born series expansion of the scattering, and when the sample thickness

becomes comparable to the MFP, the higher-order terms become comparable. Supplementary Fig. 9 shows the relative contributions in different STEM imaging modes where scattering outside the aperture is collected as the annular dark field image, as well as the incoherent BF image. As the image is defocused, and the incoherent imaging modes do not benefit from tilt correction, they are blurred into a slowly varying background while the tcBF contrast remains sharp.

Compared to ptychography, we find tcBF will still produce a robust image under thickness and dose conditions where our current ptychographic algorithms fail to converge, and indeed there is a benefit to initializing a ptychographic reconstruction with the information provided by tcBF[26], especially the estimate of the probe shape. Furthermore, at low doses where the signal is dominated by the central disk, our analysis summarized in Fig. 5b,c gives insight into where the signals accessible to ptychography are encoded. Close to in-focus conditions, only the anti-Friedel term of the PCTF used in DPC and SSB imaging is available. At large defocus, the Friedel term of the PCTF, used in tcBF and providing phase contrast at low frequencies, is also accessible. This suggests low-dose ptychography should be performed at large defocus conditions similar to those used for tcBF.

Overall, there are many lessons learned in the decades it took for EFTEM SPA to reach its present resolution that can also be applied to both the algorithm development and experimental design for tcBF and ptychography to boost their performance to comparable levels for thin specimens, and potentially well beyond for thick samples. Three key differences that are applied in EFTEM, which have not yet been applied in STEM, are corrections for beam-induced motion and damage by (i) movie-mode acquisition and (ii) dose fractionation in EFTEM, and (iii) corrections for image distortion in (tcBF-)STEM due to sample drift. These could in principle be applied iteratively, with optimization based on the resulting image resolution metric.

The movie mode and motion correction developed for conventional TEM operation mode effectively accounts for the thermal and mechanical drift and the beam-induced specimen motion[47,48]. In tcBF-STEM, upsampling effectively reduces the data acquisition time, thereby mitigating the impact of slow drift but beam-induced motion is not yet corrected. Beam-induced motion, reflected by the large B factors in our tcBF reconstruction and other contemporary ptychography reconstructions, may well be a major factor limiting resolution. Current 4D-STEM pixel array detectors are still too slow to incorporate these corrections directly, but analogous correction modes should be possible: both tcBF and ptychography already contain information in the overlapping probe positions that could be used to correct the beam-induced motion. At present this correction is limited by the dose per recorded diffraction pattern, but a fast detector design with a larger pixel count could address this by allowing for a larger illuminated area/pattern.

In summary, to fully exploit this information may require a new, faster generation of detectors and scan systems to meaningfully decipher the underlying specimen motion and time-ordered information. While the tcBF-STEM in this study was slower than EFTEM in terms of data acquisition, this limitation is due to the detector used here. There are now existing STEM detectors operating at 100 kHz, which is 10–200 times faster than the detectors used here, enabling comparable speeds to EFTEM in terms of total acquisition time. Within a year, commercial event-driven detectors are expected to boost speeds by another factor of 10, potentially making STEM collection faster.

Future efforts aimed to enhance the performance of tcBF-STEM would involve addressing beam-induced specimen motion and exploring the practical resolution limits of this technique. This includes using an increased probe convergence angle and higher-order probe aberrations, as well as exploiting the parallax effect to determine and correct the defocus for individual structures within thick sections.

## Online content

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

## Methods

### tcBF-STEM upsampling

Real-space sampling of the tcBF data collection is closely linked to defocus and convergence angles. Here we discuss how to design a sampling strategy for tcBF. We start with diffraction patterns from a spot size of width $\Delta w$ that are recorded a distance $\Delta d$ apart. For a convergence angle $\alpha$ and defocus $\Delta f$, the images formed from the off-axis pixels on the detector will produce shifted images that cover a range of displacements over $\pm (\alpha \, \Delta f)$. To ensure no gaps in sampling, the step size should be smaller than $\Delta d = \sqrt{2}\alpha \, \Delta f$ (Extended Data Fig. 2). This still leads to a nonuniform overlap and dose distribution, so in general the step size should be kept smaller than $\Delta d = \alpha \, \Delta f$, with the dose distribution becoming more uniform as the step size decreases (with the trade-off of increased acquisition time). As the image reconstruction algorithm is a linear process, the dose can be distributed over multiple overlapping regions. The time ordering of this acquisition could also allow for a movie-mode-like reconstruction analysis as a function of dose. For a detector pixel of width $\Delta \alpha$ (or inelastic blur $\Delta \alpha$), the blur across a single pixel is $\delta = \Delta \alpha \, \Delta f$ so the largest useful upsampling ratio is roughly $\frac{\Delta w}{\delta} = \frac{\alpha}{\Delta \alpha}$.

As an example, for the dataset shown in Fig. 2, scan step size ($\Delta d$) is 8 Å and $\alpha$ is 5.5 mrad. With $\Delta f$ = 1.3 μm applied, $\Delta w$ is about 13 nm. With $\Delta d$ substantially smaller than $\alpha \, \Delta f$, the collected diffraction patterns contain a substantial amount of overlapping information. This surplus of information is utilized to achieve real-space upsampling through sub-(scan)-pixel image shifting. To implement upsampling, each image formed by a single detector pixel is upsampled by Fourier transforming, padded with zeroes and inverse transformed, before shift correction (Supplementary Fig. 1) and summing. The combined image is then weighted by the distribution of sub-pixel image shifts (Supplementary Fig. 2). Different padding options are also compared and assessed in Supplementary Fig. 3. Zero padding is observed to preserve information transfer beyond the scan sampling. The PCTF simulation for tcBF and BF uses 5.5 mrad for convergence semi-angle and 700-nm defocus. Measuring image shifts in tcBF-STEM can also be regularized using the probe aberration function. All cryogenic tcBF images presented here benefit from this regularization, and a comparison with and without regularization is shown in Supplementary Fig. 10.

The limits of upsampling practically depend on several factors in addition to the optical resolution limit. Reciprocal-space sampling, real-space probe overlapping and real-space image shift accuracy are critical factors for information retrieval through upsampling. Reciprocal-space sampling is primarily determined by the camera length, which is chosen to optimize the collection angle and angular resolution for a given detector. The degree of upsampling we can achieve is also limited by the accuracy of the image shift determination. Insufficient SNR in cross-correlation can hinder the accuracy of image shift determination, which usually happens when the image SNR is low. It is possible to improve the accuracy by leveraging knowledge of the expected probe aberration function. It is also important to note that there is a trade-off between fineness of the reciprocal-space sampling and the SNR in the images formed by individual pixels in real space. Additionally, variations in the CTFs from higher-order aberrations and the impact of the aperture edge at different angular positions in the diffraction space, can also lead to false shift determinations.

### Comparative analysis on thick samples

The organelles shown in Fig. 3, Extended Data Fig. 7 and Supplementary Fig. 4 were isolated and purified from the HEK293T cells. Cells were mechanically lysed by osmotic shock and needle shearing[49]. The 4D-STEM data were recorded using an EMPAD[19] on a TFS Krios G4 with a 7-mrad semi-convergence angle and a 2.8-nm scan step size. Then, 256 × 256 scan positions were collected. The corresponding EFTEM images were recorded using a Falcon 4i detector and the Selectris X

energy filter with a slit width of 10 eV on the same TFS Krios G4. Acceleration voltage was 300 kV and the spherical aberration of the objective lens was 2.7 mm. tcBF images are reconstructed with the iterative alignment provided in py4DSTEM[32] and upsampling is implemented in an in-house Python package based on the method described in the upsampling section. For the EFTEM images, 57 frames were recorded, motion corrected using MotionCor2 (ref. 48) and the CTF in Table 1 is estimated with CTFFIND4 (ref. 43).

For Supplementary Fig. 5, the *E. coli* specimens were prepared from the GL002 strain and plunge frozen with 200-mesh Quantifoil 2/1 holey carbon copper TEM grids. The images were recorded on a customized Thermo Scientific Titan Themis with a Gatan 626 cryo-transfer holder at 300 kV. The 4D-STEM data were recorded using an EMPAD[19] with a 2-mrad convergence angle. tcBF images were reconstructed with an in-house Python package where the alignment algorithm is based on rigid shift registration between every possible pair[50] and upsampling algorithm as described. The EFTEM images were acquired with a K2 Summit direct detector (Gatan) operating in linear mode. For all the EFTEM images, short exposures were collected in the movie mode and cross-correlated with the number of frames chosen to match the dose of the corresponding STEM image. CTF modulation can also affect the qualitative comparison. However, achieving the exact same defocus for the two techniques can be challenging because the samples are thick (~550 nm to 700 nm; Table 1) and not flat, and switching between TEM and STEM modes is a nontrivial change in optical alignments. As a result, we present a series of cases where EFTEM images are measured to have defoci larger, equivalent to or smaller than those of tcBF, alongside a scenario where both techniques are targeted at the same nominal defocus.

To check the sample damage as a function of dose, we consecutively acquire tcBF-STEM images with doses ranging from 1.5 e$^-$/Å$^2$ to 210 e$^-$/Å$^2$ (Supplementary Fig. 11). After a cumulative exposure of 280 e$^-$/Å$^2$, no obvious sign of bubbling is observed, consistent with a previous STEM study[16], while visible bubbling effects start to form after a total exposure over 150 e$^-$/Å$^2$ for conventional TEM[16]. This does not mean no damage has occurred, but rather damage products have not migrated over longer length scales.

### MFP and thickness estimates

Measurements of the inelastic MFP (scaled to 300 keV (ref. 51)) range from 100 nm for amorphous carbon to 275 nm for proteins to 310 nm for vitreous ice[52], scaling roughly with the degree of hydrogenation. For our thickness measurements, we used the inelastic MFP of ice. The elastic MFP is more strongly dependent on the range of collection angles as the elastic scattering has a much wider angular distribution than inelastic scattering. Thus, what is often reported is $n$, the ratio of elastic to inelastic scattering for a given measurement geometry, and this is in the range of 2–5, with 3 being a typical value for cryo-EM of organic systems[53], suggesting a typical elastic MFP is about 700–900 nm. We calculated the elastic MFP from a multi-slice simulation of amorphous ice, for a 50-nm- and 200-nm-thick supercell and a 5.5-mrad convergence and collection angle at 300 keV. We model thermal diffuse scattering with the frozen phonon approximation. Averaging over multiple configurations, we fit the decay of the central beam to find the elastic MFP, $\lambda_{el}$ = 774 ± 45 nm. Extended Data Fig. 10 shows the elastic scattering and elastic MFP as a function of scattering angle. The elastic MFP sets a thickness for which the dominant contrast mechanism crosses over from phase contrast to scattering absorption contrast.

In relating the signal remaining in an energy-filtered image, $I_{EFTEM}(t) = I_{TEM} \exp(-\frac{t}{\lambda_{in}})$ and $I_{TEM}$ is the corresponding unfiltered image. Even when no objective aperture is used, there is still some high-angle elastic scattering (including backscattering) that does not reach the detector, so not all of the incident beam is collected and $I_{TEM}(t) = I_0 \exp(-\frac{t}{\lambda_{HA}})$. Combining these results, we get equation (1):

$$I_{\text{EFTEM}}(t) = I_0 \exp\left(-\frac{t}{\lambda_{\text{HA}}}\right) \exp\left(-\frac{t}{\lambda_{\text{in}}}\right) = I_0 \exp\left(-\frac{t}{\lambda_{\text{in}'}}\right) \tag{1}$$

For our microscope, we measured $\lambda_{\text{HA}} = (46 \pm 1)\lambda_{\text{in}}$, and it is convenient to keep the functional form $\lambda_{\text{HA}} = \alpha\lambda_{\text{in}}$. From equation (17) we can calculate the high angle correction to the inelastic MFP as equation (2):

$$\lambda_{\text{in}'} = \frac{\alpha}{(\alpha+1)}\lambda_{\text{in}} \tag{2}$$

so $\lambda_{\text{in}'} = 0.979\lambda_{\text{in}}$ and $\lambda_{\text{in}} = 303$ nm for ice.

### Single-particle analysis on VLPs

The specimen is a coat protein of bacteriophage PP7 that self-assembled during recombinant expression in *E. coli*. R1.2/1.3 mesh 300 UltrAuFoil TEM grids were used. STEM images were acquired on the customized Thermo Scientific Titan Themis with a Gatan 626 cryo-transfer holder. Images were recorded with 300-kV acceleration voltage on an EMPAD-G2 detector[20], with an 8-mrad semi-convergence angle, an 11-Å scan step size and a 45 e⁻/Å² total exposure dose, and upsampled to an image pixel size of 2.77 Å. A typical scan size is 512 × 512 with a 100-µs dwell time. tcBF images are reconstructed with the iterative alignment provided in py4DSTEM[32] and upsampling is implemented in an in-house Python package based on the method described in the upsampling section. The SPA reconstruction is obtained with cryoSPARC[42] where CTFFIND4 (ref. [43]) is used to estimate global CTF. Particles are picked with a template generated by manually picked particles. The final 3D reconstruction has icosahedral symmetry and a dynamic mask imposed.

The comparative analysis of the VLPs with EFTEM is performed with a Cs-corrected TFS Krios, shown in Extended Data Fig. 9. The EFTEM data were acquired at an accelerating voltage of 300 kV, a pixel size of 1.076 Å and the total dose is 52.18 e⁻/Å². The defocus ranges from −1 µm to −2 µm. The final reconstruction is obtained with 900 particles, and the analysis is also done with cryoSPARC[43].

### The CTF for tilted-beam imaging

Considerable insight into the imaging behavior can be obtained from the linear imaging theory, even though this simple approximation must break down under multiple scattering in thick samples, where numerical simulation is needed for quantitative modeling. The goal here is to provide the same level of approximation for STEM as that which is widely used as a starting point in EFTEM analysis. In linear imaging theory the image contrast $C(\omega)$ can be written as equation (3):

$$C(\omega) = \text{PCTF}(\omega)\frac{F_p(\omega)}{\lambda} \tag{3}$$

where $F_p$ is the elastic scattering amplitude of the projected object and $\lambda$ is the electron wavelength[48]. In general, the PCTF can be complex, with the real part corresponding to angularly symmetric (that is, Friedel-like) scattering, and the imaginary part to antisymmetric scattering (at the lowest order of approximation, these terms would correspond to weak phase and weak amplitude approximations). Rose considered the phase contrast for samples which have undergone both elastic and inelastic scattering, with the case for a tilted beam given by equation 26 of ref. [48]. For weakly scattering objects, the quadratic and higher-order terms in his equation (26) can be neglected and a simpler, linear PCTF is given by Rose's equation (4):

$$\text{PCTF}(\boldsymbol{\omega}) =$$
$$\frac{i}{\Omega_0}\int A(\Theta)D(\Theta)\{A(\boldsymbol{\omega}-\Theta)e^{-i[\chi(\boldsymbol{\omega}-\Theta)-\chi(\Theta)]} - A(\boldsymbol{\omega}+\Theta)e^{+i[\chi(\boldsymbol{\omega}+\Theta)-\chi(\Theta)]}\}d^2\Theta \tag{4}$$

which we interpret here in terms of the STEM geometry where $\boldsymbol{\omega}, \Theta$ are momentum vectors projected onto the detector in the diffraction plane and normalized as scattering angles (which are a vector in this plane, hence the bold notation). We also introduce the factor of ½ to be consistent with the modern definition that the magnitude of the PCTF is ≤ 1 (see ref. [48]; 2nd column, top of page 259). $A(\Theta)$ and $D(\Theta)$ are the probe-forming and detector functions, which are 1 inside the apertures, and 0 outside. $\chi(\boldsymbol{\omega})$ is the aberration function of the objective lens and $\Omega_0 \approx \pi\alpha^2$ is the solid angle subtended by the objective aperture, which cuts off at angle $\alpha$. For a pixelated detector with small pixels (that is, the change in $\chi$ across a single pixel is small), $D(\Theta) \approx \delta(\Theta)$ and equation (4) simplifies to equation (5):

$$\text{PCTF}(\boldsymbol{\omega},\Theta) = \frac{i}{\Omega_0}A(\Theta)\{A(\boldsymbol{\omega}-\Theta)e^{-i[\chi(\boldsymbol{\omega}-\Theta)-\chi(\Theta)]} - A(\boldsymbol{\omega}+\Theta)e^{+i[\chi(\boldsymbol{\omega}+\Theta)-\chi(\Theta)]}\} \tag{5}$$

where $\omega$ is the spatial frequency in the image, and $\Theta$ is the collection angle (that is, pixel) on the detector, so PCTF $(\boldsymbol{\omega},\Theta)$ gives the PCTF for the image formed by scanning the probe in the sample plane and collecting the signal at pixel $\Theta$ on the detector. The PCTF $(\boldsymbol{\omega},\Theta)$ without an aperture is shown in Extended Data Fig. 3 for a range of different tilts $\Theta$, and the corresponding PCTF $(\boldsymbol{\omega},\Theta)$ for an aperture is shown in Extended Data Fig. 4.

For the special case of axial illumination ($\Theta = 0$), the PCTF reduces to the BF CTF of $-\sin(\chi(\omega))$. This would also have a cutoff at $|\boldsymbol{\omega}| = \alpha$. The tilted-beam case has nonzero contributions outside the aperture, up to a cutoff of $2\alpha$ when $|\Theta| = \alpha$ from the terms $A(\Theta)A(\boldsymbol{\omega}-\Theta)$, $A(\Theta)A(\boldsymbol{\omega}+\Theta)$. This is the same information limit as the annular dark-field and iDPC imaging, and double that of the axial BF signal. The power spectrum of the apertured PCTF (Extended Data Fig. 4) shows the double-resolution limit.

We can get a sense of how the aberrations lead to shifts in the image, by considering the special case where the dominant aberration is defocus so (equation (6)):

$$\chi(\Theta) = -1/2k_0\Delta f|\Theta|^2 \tag{6}$$

where $k_0 = \frac{2\pi}{\lambda}$. The PCTF then further simplifies to equation (7):

$$\text{PCTF}(\boldsymbol{\omega},\Theta)$$
$$= \frac{i}{\Omega_0}A(\Theta)\{A(\boldsymbol{\omega}-\Theta)e^{+1/2ik_0\Delta f\omega^2} - A(\boldsymbol{\omega}+\Theta)e^{-1/2ik_0\Delta f\omega^2}\}e^{-i(\Delta f\Theta)\cdot(k_0\boldsymbol{\omega})} \tag{7}$$

From the Fourier shift theorem, when transforming from the diffraction plane $k_0\boldsymbol{\omega}$ to the image plane $\mathbf{x}$, the $e^{-i(\Delta f\Theta)\cdot(k_0\boldsymbol{\omega})}$ term in equation (7) gives a shift of the image in real space of $\Delta f\Theta$, that is a shift proportional to the defocus and the angle from the axis on the detector. This is the tilt that is corrected by tcBF. The defocus aberration from the $e^{\pm 1/2ik_0\Delta f\omega^2}$ terms are still present in the CTF and the tilt-corrected PCTF becomes equation (8):

$$\text{PCTF}(\boldsymbol{\omega},\Theta) = \frac{i}{\Omega_0}A(\Theta)\{A(\boldsymbol{\omega}-\Theta)e^{+1/2ik_0\Delta f\omega^2} - A(\boldsymbol{\omega}+\Theta)e^{-1/2ik_0\Delta f\omega^2}\} \tag{8}$$

The tcBF CTF is obtained by summing over all tilt angles $\Theta$. This is most easily accomplished by first summing symmetrically over pairs of angles at $\Theta$ and $-\Theta$ (equation (9)):

$$\text{PCTF}(\boldsymbol{\omega},+\Theta) + \text{PCTF}(\boldsymbol{\omega},-\Theta)$$
$$= \left(\frac{-2}{\Omega_0}A(\Theta)\{A(\boldsymbol{\omega}-\Theta)+A(\boldsymbol{\omega}+\Theta)\}\sin\left(\frac{1}{2}k_0\Delta f\omega^2\right)\right) \tag{9}$$

and then completing the sum over half of the central disk (say all $\Theta_x > 0$). In polar coordinates $\Theta = (\theta, \phi)$ and for a disk of diameter $\alpha$, we integrate over $\theta$ and $\phi$ (equation (10)):

$$\text{CTF}_{\text{tcBF}}(\omega) = -\left(\frac{2}{\pi\alpha^2}\right)\left[\int_0^\pi d\phi \int_0^\alpha \Theta d\Theta A(\Theta)\{A(\omega - \Theta)\right. \tag{10}$$
$$\left. + A(\omega + \Theta)\}\right]\sin\left(\frac{1}{2}k_0\Delta f\omega^2\right)$$

The integral over $\Theta$ gives the area of the overlap of disks of diameter $\alpha$ that are $\omega$ apart, and can be found in the appendix of ref. 48 as equation (11):

$$\mathcal{L}(\omega) = \begin{cases} \frac{2}{\pi}\left[\cos^{-1}\left(\frac{1}{2}\omega\right) - \frac{1}{2}\sqrt{1 - \frac{1}{4}\omega^2}\right], 0 \leq \omega \leq 2 \\ 0, \omega \geq 2 \end{cases} \tag{11}$$

$\mathcal{L}(\omega)$ is the well-known envelope for a self-luminous object, such as for the annular dark field CTF. The tcBF CTF can then be written more compactly as equation (12):

$$\text{CTF}_{\text{tcBF}}(\omega) = -\mathcal{L}(\omega)\sin\left(1/2k_0\Delta f\omega^2\right) \tag{12}$$

## Comparison of the CTF for tcBF with DPC

The optimal CTF for DPC and iDPC is the in-focus condition with no aberrations inside the aperture. Then $\chi(\Theta) = 0$ and the general PCTF simplifies to equation (13):

$$\text{PCTF}(\omega) = \left(\frac{i}{\Omega_0}\right)A(\Theta)\{A(\omega - \Theta) - A(\omega + \Theta)\} \tag{13}$$

that is, $\text{Re}(\text{PCTF}) = 0$ at zero defocus, and only the antisymmetric component remains (Extended Data Fig. 4c). The $\text{DPC}_x$ signal is produced by subtracting all the left-tilted ($\Theta_x < 0$) from the right-tilted ($\Theta_x > 0$) detector signals and then summing to produce the DPC CTF of Fig. 5b.

At $\Theta = 0$ the $\text{PCTF}(\omega) = 0$, and the PCTF remains 0 so long as $|\omega| < \alpha$, $|\omega - \Theta| < \alpha$ and $|\omega + \Theta| < \alpha$, giving the white regions in each frame of Extended Data Fig. 4c. In ptychography, this is referred to as the triple-overlap region[46], reflecting the simultaneous overlap of the $+\omega$ and $-\omega$ beams with the incident beam (in ptychography, this is usually displayed in detector plane $\Theta$ for a range of selected $\omega$, while we have displayed the $\omega$ plane for a range of selected $\Theta$), and is zero for in-focus imaging. When a phase shift is deliberately introduced, this triple overlap provides the phase contrast for BF imaging, but still remains zero for DPC and SSB ptychography (Extended Data Fig. 4b). DPC and SSB rely on the double-overlap region where $|\omega| < \alpha$ and either $|\omega - \Theta| < \alpha$ or $|\omega + \Theta| < \alpha$, but not both. Again, the information limit is the largest value of $\omega$ for which the PCTF is not zero. This occurs at $|\Theta| = \alpha$ and $\omega = 2\Theta$, so the largest nonzero value of $|\omega|$ is $\omega = 2\alpha$, double the radius of the aperture.

It is important to note that the CTF in the triple-overlap region has double the amplitude of that of the double-overlap region (Fig. 3 of ref. 46). This suggests that tcBF should have the potential to reach double the dose efficiency of DPC at below spatial frequencies where $|\omega| < \alpha$ and a $\frac{\pi}{2}$ phase shift can be introduced through the aberration function. This difference becomes very noticeable at low spatial frequencies where the double-overlap terms tend to be zero, and the triple-overlap contrast can be boosted by increasing defocus.

## Comparison of the DQE for tcBF and iDPC

The iDPC CTF is obtained from the DPC CTF by integration in real space, corresponding to a division by spatial frequency in Fourier space as equation (14):

$$\text{PCTF}_{\text{iDPC}}(\omega) = \frac{\text{PCTF}_{\text{DPCx}}(\omega) + i\text{PCTF}_{\text{DPCy}}(\omega)}{i(k_x + ik_y)} \tag{14}$$

The power spectrum of the recorded DPC image in the presence of a noise spectrum $N(\omega)$ is given by equation (15):

$$P_{\text{DPC}}(\omega) = |\text{PCTF}_{\text{DPCx}}(\omega)|^2 \frac{|F_p(\omega)|^2}{\lambda^2} + \alpha^2 |N(\omega)|^2 \tag{15}$$

The power spectrum for iDPC based on the DPC measurement with noise is given by equation (16):

$$P_{\text{iDPC}}(\omega) = \frac{|\text{PCTF}_{\text{DPCx}}(\omega)|^2 + |\text{PCTF}_{\text{DPCy}}(\omega)|^2}{(k_x^2 + k_y^2)}\frac{|F_p(\omega)|^2}{\lambda^2} + \alpha^2 \frac{|N_x(\omega)|^2}{(k_x^2 + k_y^2)} \tag{16}$$

The DQE of the measurement with a noise power spectrum, NPS $(\omega)$, is given by equation (17):

$$\text{DQE}(\omega) = DQE(0)\frac{|PCTF(\omega)|^2}{|NPS(\omega)|^2} \tag{17}$$

By convention, the noise power spectrum is normalized to 1, and collection efficiency is absorbed into the DQE $(0)$[54]. For an ideal detector pixel, $\text{DQE}(0) = 1$ and for DPC imaging the DQE $(\omega)$ becomes equation (18):

$$\text{DQE}_{\text{DPCx}}(\omega) = \frac{|\text{PCTF}_{\text{DPCx}}(\omega)|^2}{\alpha^2|N(\omega)|^2} \tag{18}$$

and after integrating, the DQE for iDPC becomes equation (19):

$$\text{DQE}_{\text{iDPC}}(\omega) = \frac{|\text{PCTF}_{\text{DPCx}}(\omega)|^2 + |\text{PCTF}_{\text{DPCy}}(\omega)|^2}{2\alpha^2|N(\omega)|^2} \tag{19}$$

This has a very similar shape to the DPC DQE because the noise is amplified in the same way as the signal.

Similarly, applying equation (13) for tcBF, we find the tcBF DQE to be equation (20):

$$\text{DQE}_{\text{tcBF}}(\omega) = \frac{|\text{PCTF}_{\text{tcBF}}(\omega)|^2}{\alpha^2|N(\omega)|^2} \tag{20}$$

For an ideal detector the noise spectrum is only from Poisson noise, which is flat, so the differences in DQE for tcBF and iDPC can be understood by comparing the squares of the PCTFs for tcBF and DPC (not iDPC; Fig. 5b,c). As a consequence, iDPC has a poor DQE at low spatial frequencies compared to defocused tcBF.

### Reporting summary

Further information on research design is available in the Nature Portfolio Reporting Summary linked to this article.

## Data availability

The 4D-STEM datasets for Fig. 3, Extended Data Fig. 7 and Supplementary Fig. 4, along with the corresponding EFTEM images acquired in the same regions, are publicly accessible on Zenodo at https://doi.org/10.5281/zenodo.10825338 (ref. 55). The primary map, half maps, FSC and mask used for the reported FSC in Fig. 4 have been deposited on the Electron Microscopy Data Bank under accession EMD-48236. The same for Extended Data Fig. 9 are deposited under EMD-70328. The model docked in the map shown in Fig. 4 is available from the PDB under accession code PDB 1DWN. Source data are provided with this paper.

## Code availability

Python packages for tcBF-STEM are available on GitHub at https://github.com/yyu2017/tcBFSTEM/. Additionally, example data and

notebooks are accessible on Code Ocean via https://doi.org/10.24433/CO.4000019.v1.

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

## Acknowledgements

This work is supported by the NSF (DMR-1654596, DMR-1429155, DMR-1719875, DMR-2039380), the Packard Foundation and Chan Zuckerberg Institute for Advanced Biological Imaging. This work made use of the instruments at Chan Zuckerberg Institute for Advanced Biological Imaging, the Cornell Center for Materials Research (CCMR) Shared Facilities and PARADIM. CCMR facilities and X.S.Z. are supported through the NSF MRSEC program (DMR-1719875). PARADIM and S.E.Z. are supported by the NSF MIP program (DMR-2039380). We are grateful for all the time that L.F.K. was able to share with us. May her memory be a blessing. We thank T. Wang for inspiring discussions on tcBF upsampling. We appreciate Y. Xu for providing the *E. coli* specimens, and M. D. Leonetti's group for providing the cell lines for the organelle specimens. In addition, we thank E. J. Kirkland for helpful discussions on tilted BF CTFs and P. Cueva (NSF PHY-1549132) for help with aberration and tilt measurements in 4D-STEM. We acknowledge G. Varnavides, S. M. Ribet and C. Ophus for helpful discussions on improving algorithms for tcBF. Finally, we thank B. Carragher, C. S. Potter and D. Agard for advice on experimental designs for comparing tcBF-STEM to EFTEM, as well as for insights on future steps to improve the technique.

## Author contributions

Y.Y., K.A.S., D.A.M. and L.F.K. designed the tcBF experiments. Y.Y, K.A.S. and K.X.N. performed the tcBF experiments. Y.Y., K.A.S., M.C., K.X.N, D.A.M. and L.F.K. developed the tcBF algorithm and analyzed the tcBF data. S.E.Z. and D.A.M. calculated the tcBF CTFs. M.P., D.S. and H.S. prepared the purified cellular organelle samples. Y.Y., M.P., M.K. and C.D. analyzed the single-particle data. Y.Y., K.A.S., L.F.K. and D.A.M. wrote the paper with input from all authors.

## Competing interests

The authors declare no competing interests.

## Additional information

**Extended data** is available for this paper at https://doi.org/10.1038/s41592-025-02834-9.

**Correspondence and requests for materials** should be addressed to Yue Yu or David A. Muller.

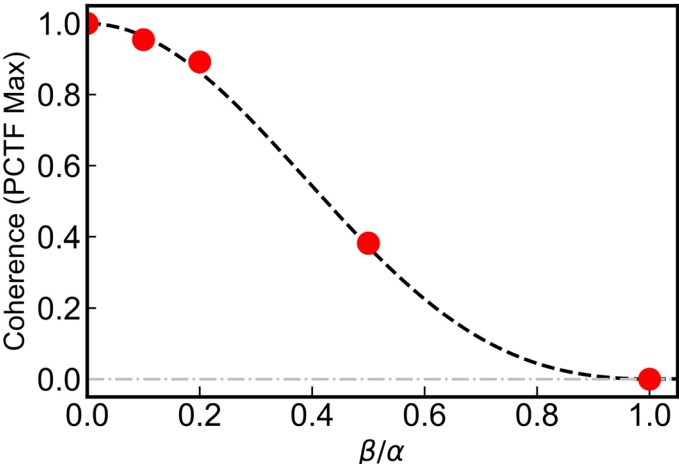

**Extended Data Fig. 1 | Degree of partial coherence as function of increasing collection angle $\beta$ with respect to the probe-forming convergence angle $\alpha$.** Angles $\alpha$, $\beta$ for tcBF re shown in the schematic in Fig. 1b. The red dots are the maximum value in the phase contrast transfer function at Scherzer defocus (see Kirkland Chapter 3.4 for details), and the black dashed curve is $|j_{12}|^2$ where the complex degree of coherence $j_{12} = (2J_1(v)/v)e^{i\psi}$ (eqn 28 of Born and Wolf,

Chap 10.4). The coherence drops to zero when the collection angle is equal to the convergence angle. It does have very small but decaying secondary maxima at larger angles. The peak coherence is reduced by 20% when $\beta/\alpha = 0.25$ so to tolerate a 1 mrad blur at the detector (for example from inelastic scattering), the probe-forming aperture should be kept larger than 4 mrad to preserve a strong coherence.

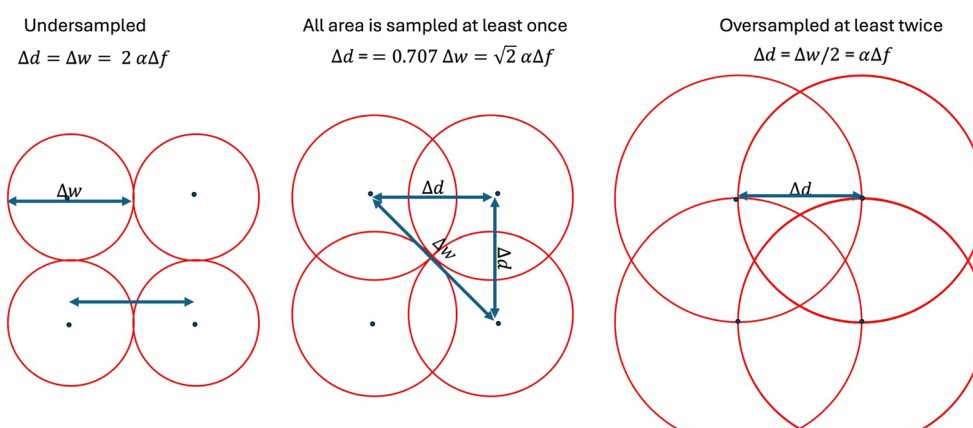

**Extended Data Fig. 2 | Efficient Sampling with a defocused probe.** Diffraction patterns from a spot size of width $\Delta w$ are recorded a distance $\Delta d$ apart. For a convergence angle $\alpha$ and defocus $\Delta f$, the images formed from the off-axis pixels on the detector will produce shifted images that cover a range of displacements over $\pm (\alpha \Delta f)$ so $\Delta w = 2(\alpha \Delta f)$. **(left)** At a sampling $\Delta w = \Delta d$, there are regions of the sample that are not exposed to the electron beam so the image is undersampled and information can be missed. **(middle)** To ensure no gaps in sampling, the step size should be smaller than $\Delta d = \sqrt{2}\alpha \Delta f$. This still leads to a non-uniform overlap and dose distribution. **(right)** In general, the step size should be kept smaller than $\Delta d = \alpha \Delta f$, with the dose distribution becoming more uniform as the step size decreases (with the tradeoff of increased acquisition time). As the image reconstruction algorithm is a linear process, the dose can be distributed over multiple overlapping regions. The time-ordering of this acquisition could also allow for a movie-mode-like reconstruction analysis as a function of dose.

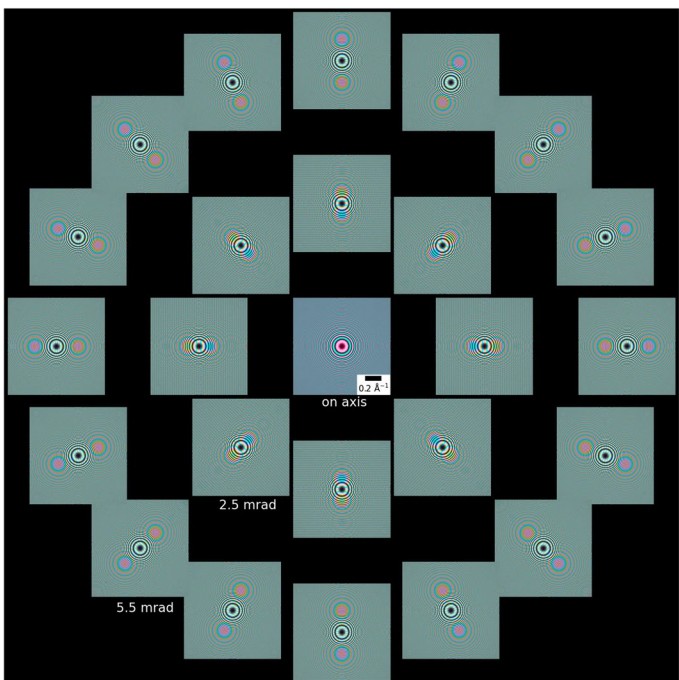

**Extended Data Fig. 3 | Aperture-free, complex PCTF in a Zemlin tilt-tableau.**
The aperture-free, complex PCTF for 300 kV electrons and 700 nm defocus in a Zemlin tilt-tableau out to 5.5 mrad of tilt. The x-y coordinates within each frame are spatial frequencies of the image, $\omega$, and the tilt offset of each frame is $\Theta$. With no objective (condenser) aperture and no higher order aberrations, the power spectrum of each image under tilted illumination is identical.

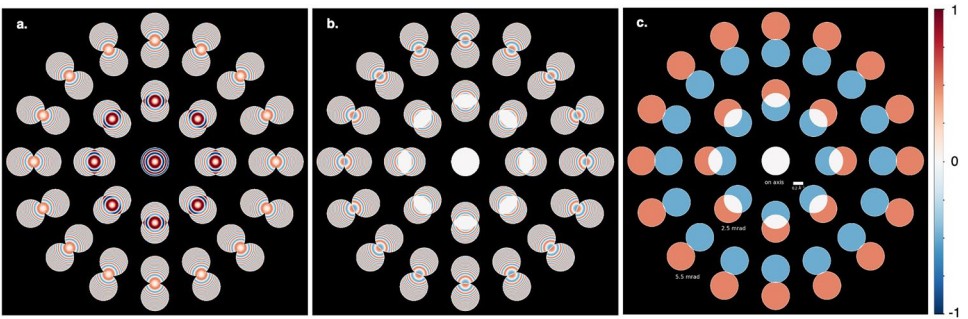

**Extended Data Fig. 4 | The symmetric and antisymmetric components of the PCTF with a 5.5 mrad objective (condenser for STEM) in a Zemlin tilt-tableau.** The symmetric and antisymmetric components of the PCTF for 300 kV electrons with a 5.5 mrad objective (condenser for STEM) in a Zemlin tilt-tableau out to 5.5 mrad of tilt. (**a**) $\Re e\,(PCTF)$ at 700 nm defocus showing the symmetric, Friedel term, (**b**) $\Im m\,(PCTF)$ at 700 nm defocus showing the anti-symmetric, anti-Friedel term. We have shifted the individual images which corrected the tilt-induced phase ramp, and subsequently summing the tilt-corrected images gives the PCTF shown in Fig. 2f. (**c**) $\Im m\,(PCTF)$ at zero defocus, again showing its anti-symmetric nature. The DPC-x image is formed by subtracting the left tilts from the right.

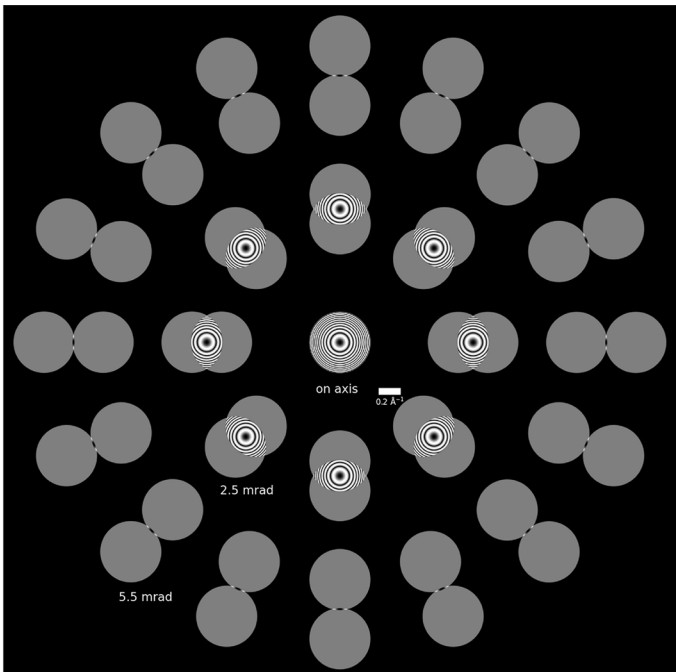

**Extended Data Fig. 5 | The power spectrum for 300 kV electrons and 700 nm defocus with a 5.5 mrad objective (condenser) in a Zemlin tilt-tableau.** The power spectrum for 300 kV electrons and 700 nm defocus with a 5.5 mrad objective (condenser) in a Zemlin tilt-tableau out to 5.5 mrad of tilt. This highlights the strong modulations in the overlap region, and the weaker transfer in the sidelobes, but with an information limit of twice the aperture radius.

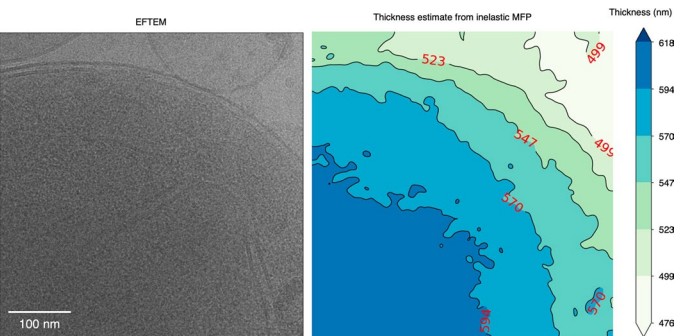

**Extended Data Fig. 6 | The thickness contour.** A contour plot of the thickness estimate for the sample shown in Fig. 3a. The thickness is estimated with the inelastic MFP using Beer's law. Using the EFTEM dataset, we obtain a ratio of $I_0/I$, where $I_0$ is the intensity recorded over vacuum, and I is the energy filtered intensity with a 10-eV slit recorded over the sample. The thickness is estimated $\ln(I_0/I)$*inelastic MFP. The inelastic MFP used here is 310 nm for vitrified ice at 300 kV.

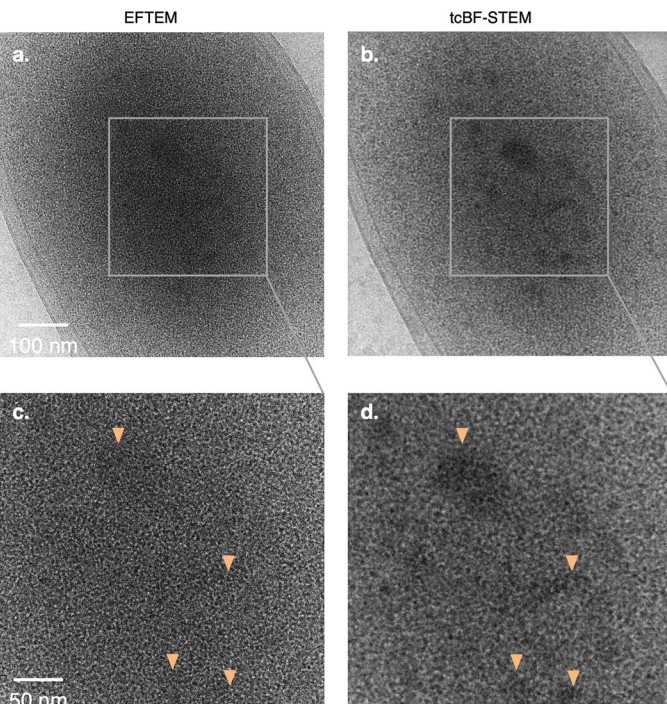

**Extended Data Fig. 7 | Comparisons of EFTEM and tcBF-STEM with different doses, defoci and acquisition orders.** Intact *E.coli* cell imaged with (**a**) EFTEM and (**b**) tcBF. With tcBF (**d**), features in the interior region of the cell are effectively resolved, whereas in EFTEM (**c**), although the same features are discernible, they are also less visible. For each comparison, the total dose measured over vacuum and the electron acceleration (300 kV) are the same, and the average thickness was estimated with the EFTEM images using the ratio of $I_0/I$ and the inelastic MFP, similar to Fig. S1. Dose is 14 $e^-/Å^2$ for both image sets. For EFTEM images, slit widths are all 10 eV and defoci are measured with CTFFIND4[43]. For tcBF images, defoci are measured with the image shifts. Defoci, thickness and other experimental details in Table 1.

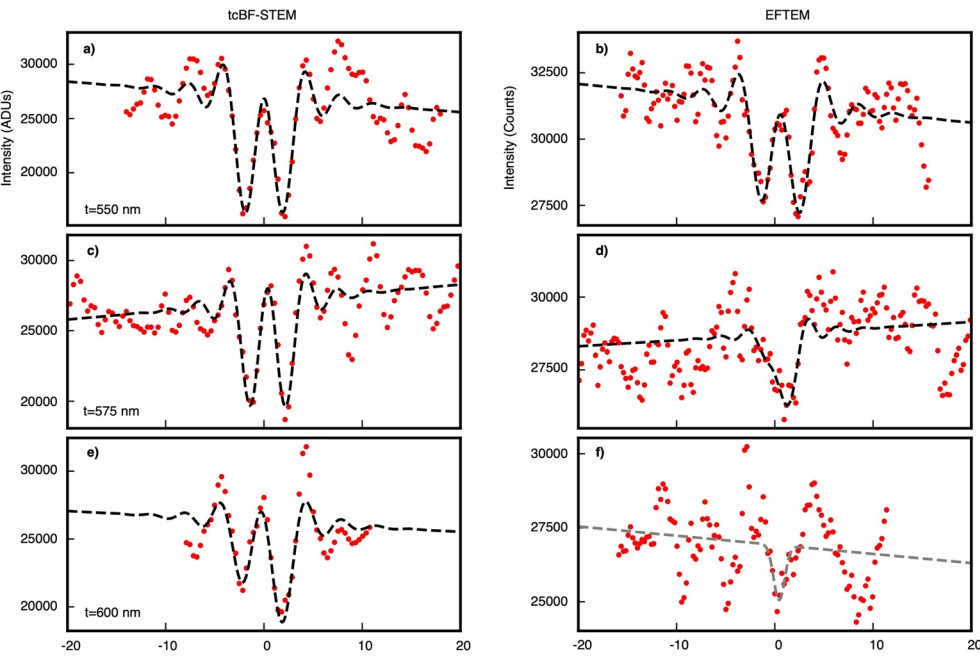

**Extended Data Fig. 8 | Contrast of the double membranes.** Measuring Contrast on the double membranes marked in Fig. 3. Left column **(a,c,e)** show the line from the tcBF image, and right column **(b,d,f)** the corresponding profiles from the EFTEM image. Local sample thickness for both is shown in the bottom left of the relevant panels. The full thickness map is shown in Fig. S9. The black dashed line is a curve fit to two sinc functions to account for the Fresnel fringes around the membrane. In the profiles from the thicker EFTEM locations **(d,f)**, the double membrane was not resolved, and the grey dashed fit to a single gaussian shows the approximate membrane location. The tcBF profiles were from 40-pixel wide line, and the EFTEM from a 60-pixel wide line to match the same dose per point in the line profile. The raw data of this figure and Fig. 3e,f are included as source data. More numerical details are included in the code that generates this figure, which is available on Zenodo[55].

**900 particles, Titan Krios, 300kV**

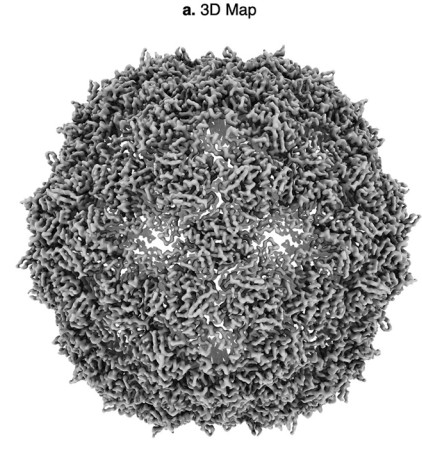

a. 3D Map

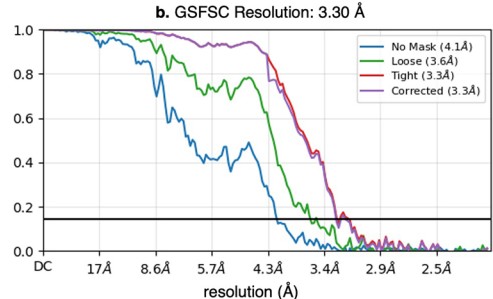

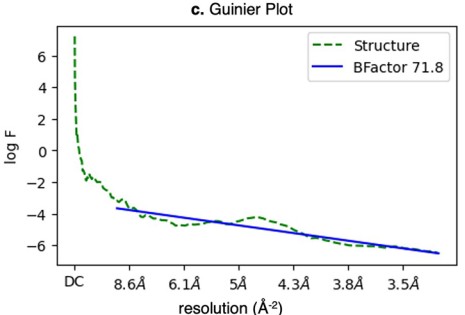

**Extended Data Fig. 9 | EFTEM of VLP PP7 SPA with 900 particles reaches a nominal resolution of 3.30 Å.** The 3D map is shown in **a**, the FSC is shown in **b** and the B factor Guinier plot is in **c**. The EFTEM data is acquired on a Cs-corrected TFS Krios at an accelerating voltage of 300 kV. The image pixel size is 1.076 Å and the total dose is 52.18 e⁻/Å². The defocus ranges from -1μm to -2μm. The final reconstruction is obtained with 900 particles and the analysis is done with cryoSPARC[44].

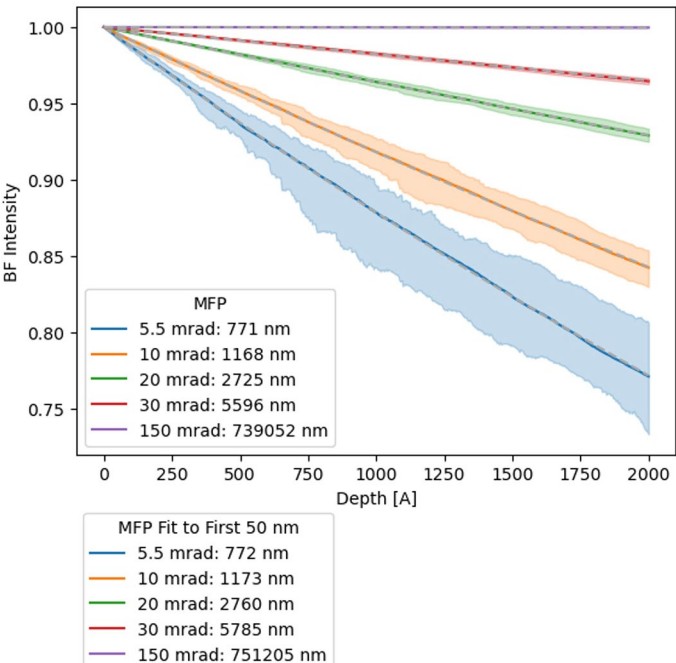

**Extended Data Fig. 10 | Elastic mean free path for a 300 kV electron in ice as a function of collection angle.** Calculate the elastic mean free path for a 300 kV electron in ice as a function of collection angle. A 200-nm thick model of amorphous ice was constructed using Kirkland's code (Chapter 7.8 of reference35) and multislice simulations for a 5.5 mrad convergence semiangle probe were performed in the frozen phonon approximation to account for thermal diffuse scattering. The elastic mean free path was fitted to simulation results, both for the 1st 50 nm (where inelastic and backscattering effects are expected to be small), and the full 200 nm thickness. Results were similar. The mean free path is defined as $\lambda = 1/n\sigma$ where $n$ is the number density of scatterers and $\sigma$ is the partial cross-section of interest, in this case elastic scattering to outside a given angle (for example the probe-forming aperture).

# Reporting Summary

## Statistics

For all statistical analyses, confirm that the following items are present in the figure legend, table legend, main text, or Methods section.

| n/a | Confirmed | |
|---|---|---|
| ☐ | ☒ | The exact sample size (*n*) for each experimental group/condition, given as a discrete number and unit of measurement |
| ☐ | ☒ | A statement on whether measurements were taken from distinct samples or whether the same sample was measured repeatedly |
| ☒ | ☐ | The statistical test(s) used AND whether they are one- or two-sided<br>*Only common tests should be described solely by name; describe more complex techniques in the Methods section.* |
| ☐ | ☒ | A description of all covariates tested |
| ☐ | ☒ | A description of any assumptions or corrections, such as tests of normality and adjustment for multiple comparisons |
| ☐ | ☒ | A full description of the statistical parameters including central tendency (e.g. means) or other basic estimates (e.g. regression coefficient) AND variation (e.g. standard deviation) or associated estimates of uncertainty (e.g. confidence intervals) |
| ☒ | ☐ | For null hypothesis testing, the test statistic (e.g. *F*, *t*, *r*) with confidence intervals, effect sizes, degrees of freedom and *P* value noted<br>*Give P values as exact values whenever suitable.* |
| ☒ | ☐ | For Bayesian analysis, information on the choice of priors and Markov chain Monte Carlo settings |
| ☒ | ☐ | For hierarchical and complex designs, identification of the appropriate level for tests and full reporting of outcomes |
| ☒ | ☐ | Estimates of effect sizes (e.g. Cohen's *d*, Pearson's *r*), indicating how they were calculated |

*Our web collection on statistics for biologists contains articles on many of the points above.*

## Software and code

Policy information about availability of computer code

| | |
|---|---|
| Data collection | EMPAD GUI 1.2.2 (2021-08-20) for 4d-stem data in figure 3 and fig 4 and supplementary fig s4 and fig s6. |
| Data analysis | Python packages for tcBF-STEM are available on GitHub at https://github.com/yyu2017/tcBFSTEM. Additionally,example data and notebook are accessible on Code Ocean DOI 10.24433/CO.4000019.v1. The code for analyzing contrast and signal-to-noise are available as source data (ED_fig8_plotlineprofiles.ipynb) Cryosparc version 4.6. |

For manuscripts utilizing custom algorithms or software that are central to the research but not yet described in published literature, software must be made available to editors and reviewers. We strongly encourage code deposition in a community repository (e.g. GitHub). See the Nature Portfolio guidelines for submitting code & software for further information.

## Data

Policy information about availability of data

All manuscripts must include a data availability statement. This statement should provide the following information, where applicable:
- Accession codes, unique identifiers, or web links for publicly available datasets
- A description of any restrictions on data availability
- For clinical datasets or third party data, please ensure that the statement adheres to our policy

The 4D-STEM datasets for Figures 3, Ext. Fig 7, Suppl. Fig. S4, along with the corresponding EFTEM images acquired in the same regions, are publicly accessible on Zenodo55. The primary map, half maps, FSC, and mask used for the reported FSC in Fig.4 have been deposited on Electron Microscopy Data Bank (EMDB) under ID

EMD-48236. The same for Ext. Data Fig. 9 is deposited under EMDB- 70328. The model docked in the map shown in Fig.4 is available from the Protein Data Bank (PDB) under accession code PDB_1DWN. The raw data of the lineprofiles in Fig.3e-f, and Ext. Data Fig.8, are provided in the source data.

# Research involving human participants, their data, or biological material

Policy information about studies with [human participants or human data](). See also policy information about [sex, gender (identity/presentation), and sexual orientation]() and [race, ethnicity and racism]().

| | |
|---|---|
| Reporting on sex and gender | N/A |
| Reporting on race, ethnicity, or other socially relevant groupings | N/A |
| Population characteristics | N/A |
| Recruitment | N/A |
| Ethics oversight | N/A |

Note that full information on the approval of the study protocol must also be provided in the manuscript.

# Field-specific reporting

Please select the one below that is the best fit for your research. If you are not sure, read the appropriate sections before making your selection.

☒ Life sciences ☐ Behavioural & social sciences ☐ Ecological, evolutionary & environmental sciences

For a reference copy of the document with all sections, see [nature.com/documents/nr-reporting-summary-flat.pdf]()

# Life sciences study design

All studies must disclose on these points even when the disclosure is negative.

| | |
|---|---|
| Sample size | No statistical methods were used to predetermine sample size. The sample sizes were chosen to cover a range of sample thicknesses, acquisition orders, and other relevant parameters, as defoci et al. The consistency in trends observed across these comparisons was deemed sufficient to consider the sample size adequate for the study's objectives. |
| Data exclusions | No data were excluded. |
| Replication | The trend of observing higher collection efficiency of tcBF-STEM compared to EFTEM was successfully replicated across all five comparisons. |
| Randomization | Randomization was not applied in this study as the selection of sample areas for method comparison was based on thickness. The same sample areas were imaged by both tcBF-STEM and EFTEM methods, making randomization unnecessary for this comparison. |
| Blinding | Blinding was not applicable in this study because the same sample areas were imaged by both tcBF-STEM and EFTEM methods, making it unnecessary to conceal the group assignments from the researchers or analysts. |

# Reporting for specific materials, systems and methods

We require information from authors about some types of materials, experimental systems and methods used in many studies. Here, indicate whether each material, system or method listed is relevant to your study. If you are not sure if a list item applies to your research, read the appropriate section before selecting a response.

## Materials & experimental systems

| n/a | Involved in the study |
|---|---|
| ☒ | ☐ Antibodies |
| ☐ | ☒ Eukaryotic cell lines |
| ☒ | ☐ Palaeontology and archaeology |
| ☒ | ☐ Animals and other organisms |
| ☒ | ☐ Clinical data |
| ☒ | ☐ Dual use research of concern |
| ☒ | ☐ Plants |

## Methods

| n/a | Involved in the study |
|---|---|
| ☒ | ☐ ChIP-seq |
| ☒ | ☐ Flow cytometry |
| ☒ | ☐ MRI-based neuroimaging |

# Eukaryotic cell lines

Policy information about cell lines and Sex and Gender in Research

| | |
|---|---|
| Cell line source(s) | HEK293T cells |
| Authentication | None of the cell lines were authenticated. |
| Mycoplasma contamination | Cell lines were not tested for mycoplasma contamination. |
| Commonly misidentified lines (See ICLAC register) | N/A |

# Plants

| | |
|---|---|
| Seed stocks | N/A |
| Novel plant genotypes | N/A |
| Authentication | N/A |

