## [Peer Review File · Nature Methods]

Dose-Efficient Cryo-Electron Microscopy for Thick Samples using Tilt-Corrected Scanning Transmission Electron Microscopy

Corresponding Author: Dr Yue Yu

Version 0:

Decision Letter:

25th Oct 2024

Dear Yue,

I understand that you are still drafting a response to the reviews of your Article, "Dose-Efficient Cryo-Electron Microscopy for Thick Samples using Tilt-Corrected Scanning Transmission Electron Microscopy, Demonstrated on Cells and Single Particles". However, since I will be away next week, I wanted to send you our decision letter. I will be happy to discuss your revision plan with you at a later date.

Although the reviewers find your work of considerable potential interest, they have raised a number of concerns. We are interested in the possibility of publishing your paper in Nature Methods, but would like to consider your response to these concerns before we reach a final decision on publication. We therefore invite you to revise your manuscript to address these concerns.

We are committed to providing a fair and constructive peer-review process. Do not hesitate to contact us if there are specific requests from the reviewers that you believe are technically impossible or unlikely to yield a meaningful outcome. We understand that it may not be possible at this time for you to add additional biological applications, but I would be happy to discuss a revision plan with you.

Link Redacted

We hope to receive your revised paper within 2 months. If you cannot send it within this time, please let us know. In this event, we will still be happy to reconsider your paper at a later date so long as nothing similar has been accepted for publication at Nature Methods or published elsewhere.

OPEN SCIENCE REQUIREMENTS

REPORTING SUMMARY AND EDITORIAL POLICY CHECKLISTS

DATA AVAILABILITY

All novel DNA and RNA sequencing data, protein sequences, genetic polymorphisms, linked genotype and phenotype data, gene expression data, macromolecular structures, and proteomics data must be deposited in a publicly accessible database, and accession codes and associated hyperlinks must be provided in the "Data Availability" section.

CODE AVAILABILITY

Please include a "Code Availability" subsection in the Online Methods which details how your custom code is made available. Only in rare cases (where code is not central to the main conclusions of the paper) is the statement "available upon request" allowed (and reasons should be specified).

MATERIALS AVAILABILITY

SUPPLEMENTARY PROTOCOL

To help facilitate reproducibility and uptake of your method, we ask you to prepare a step-by-step Supplementary Protocol for the method described in this paper. We [encourage authors to share their step-by-step experimental protocols](https://www.nature.com/nature-research/editorial-policies/reporting-standards#protocols) on a protocol sharing platform of their choice and report the protocol DOI in the reference list. Nature Portfolio's protocols.io is a free-to-use and open resource for protocols; protocols deposited onto protocols.io are citable and can be linked from the published article. More details can found at [protocols.io](https://www.protocols.io/help/publish-articles).

ORCID

Sincerely yours,
Allison

Allison Doerr, Ph.D.
Chief Editor
Nature Methods

Reviewers' Comments:

Reviewer #1 (Remarks to the Author):

The manuscript by Yu et al, "Dose-Efficient Cryo-Electron Microscopy for Thick Samples using Tilt-Corrected Scanning Transmission Electron Microscopy, Demonstrated on Cells and Single Particles", is a landmark contribution to the introduction of STEM techniques in life science cryo-EM. Central elements of the work have already had an impact on the field after presentation at seminars and conferences. These have been especially inspiring for the entry of phase-contrast STEM methods to life science cryo-EM. A very recent paper (<https://doi.org/10.1038/s41467-024-52403-5>), for example, uses the method and cites several of the conference abstracts. The manuscript under consideration presents the development of tilt-corrected BF imaging in detail and deserves recognition as the original reference.

The work presents two important advances for 4D STEM. First is the parallax correction to "realign" the STEM images produced by detector elements sitting at different points in the diffraction plane. The real-space shifts imposed by defocus, or in principle higher order lens aberrations, are compensated such that the images can be combined without loss of resolution. (A simple sum results in blur.) The second advance extends that observation to demonstrate image reconstruction by tcBF with strongly defocused acquisition. The defocused probe acquires in parallel over a larger area than the diffraction-limited probe diameter at the same semi-convergence angle. This permits the use of pixelated array detectors whose read-out speeds are much slower than single-element detectors such as a BF disk or annular dark field detector. Ptychography similarly uses a defocused probe, but is difficult to apply under the low-dose illumination conditions dictated by the radiation sensitivity of hydrated biological specimens of cryo-EM; and thick samples are a further complication. The great advantage of the tcBF is that the geometrical implementation is so straightforward and robust. Simply put, "it always works".

My most substantial complaint is that the whole theoretical framework presented in the online methods presumes that the dominant contrast mechanism is actually phase contrast. On the other hand, thick specimens scatter very strongly so amplitude contrast cannot really be ignored (lines 132 and 280 in the manuscript). At the elastic mean free path, which is roughly equal to the thickness of the specimens of interest, the fraction of intensity remaining unscattered is only 1/e. (Table 1 is probably “optimistic” in terms of transmission; it is not clear whether the numbers originate from the full field of view or from the dense areas of interest shown in the insets. The table should also display the convergence angle and depth of field for later reference.) Low angle scattering may interfere coherently within the cone of illumination, but beyond that, intensity is lost to the dark field. Indeed, such cellular samples produce strong contrast in annular dark field STEM. Nonetheless, it is clear that the diffraction-space shifting, i.e., tilt correction in the BF, makes a significant improvement in practice. My conclusion is that the protocol works well even beyond the realm in which it is formally derived in the manuscript, and I think this deserves comment. We can speculate that the coherent phase contrast contributes most significantly at the higher spatial frequencies, but as far as I’m aware there is no general solution to this problem.

Probably the simplest STEM method for thick specimens is the incoherent bright field (IBF), and so far it has been the method of choice for cryo-tomography by STEM (<https://doi.org/10.1021/acs.accounts.1c00279>, <https://onlinelibrary.wiley.com/doi/abs/10.1111/mmi.15213>). Integrating the remaining signal on axis is inverse to the dark field, by conservation, but it is less sensitive to multiple elastic scattering (with credit on that point to the Muller group, among others). tcBF should always be “equal to or better than” IBF, and it would be interesting to ask whether or when it is actually justifies the substantial effort. For a small convergence angle, the depth of field may be larger than the specimen thickness, for example the bacteria imaged with a semi-convergence of 2 mrad. One might argue, in such a case, that the effect of tilt correction will be marginal, but I suspect it will still be impressive. That would be an interesting comparison because the simplicity of working in focus with a fast single-pixel detector still has clear practical advantage. For a larger convergence angle, with a depth of field shorter than the sample thickness, the incoherent projection will just give a blur and the tilt correction is really unavoidable. The benefit should be an effectively smaller probe with uncompromised lateral resolution.

The manuscript goes into great detail on the comparison with iDPC. The text and derivations are very useful, but the emphasis seems a bit excessive. What I find missing, at least for consistency, is a discussion of the CTF for thick specimens. I interpret Fig 4b as simply the TEM-like CTF at a precise (but unspecified) defocus, presumably 700 nm as simulated in the supplementary figures S5-S7, which should predict the CTF for tcBF. If the sample is half a micron thick, how much does the CTF differ from top to bottom? The zeros would appear at different depths, possibly complicating the interpretation. 700 nm defocus is impractically low for the examples shown, both due to the specimen thickness and the 8-fold up-sampling. The simulation for iDPC, on the other hand, relates to in-focus imaging, so the fair comparison would be to consider a defocus spread over the same thickness.

All that said, it is far from clear to me that the PCTF construction is conceptually a good model for electron imaging through a bacterium. A simple acid-test is that the contrast should invert on passing through focus, which I think is very unlikely. Contrast will be dominated by scattering out of the bright field. Moreover, the angular spread and coherence loss from inelastic scattering are much more severe than usually considered. It is clear, on the other hand, that the geometric shifts of the pixels in the diffraction plane do compensate for optical aberrations.

In practice, thick samples are difficult to interpret not only because of the optical imaging conditions, but also because distinct specimen features “pile up” in a projection. One way to address this is with a very large convergence angle, even in ADF-STEM (<https://onlinelibrary.wiley.com/doi/epdf/10.1002/sml.202402871>). Another is by tomography, using a smaller convergence to get a nearly parallel projection. A third is actually iDPC (<https://www.sciencedirect.com/science/article/pii/S1047847722000077>), which shows impressive depth sectioning due to the maximum contrast in focus. Parallax deshifting tends to extend the depth of field (<https://www.cambridge.org/core/journals/microscopy-and-microanalysis/article/flexible-stem-with-simultaneous-phase-and-depth-contrast/131F0EAA69FD8D9665E7E685EC81B1BE#>), at least if we can extrapolate from quadrant detector data.

In summary, the work is certainly contributing at the state of the art, which is continually advancing. The conference abstract papers have already had an important impact, and we would all prefer to reference the authors more correctly. We are also painfully aware of the passing of Lena Kourkoutis. Nature Methods would do very well to publish such an insightful paper!

My remaining comments and suggestions for revision are more technical:

The life science readership will be confused by some of the notation. This begins in Fig 1, where the microscope component labels follow the dedicated STEM nomenclature. The objective aperture, for example, is the condenser C2 aperture for TEM. Since life science is presumably a target audience, this should just be made clear.

Inelastic scattering is often considered as described in the text: strongly forward-peaked and basically a perturbation on the elastic-scattering STEM signal. For the light elements of organic materials this is a problematic assumption. (<https://www.sciencedirect.com/science/article/pii/S0304399124000159>) The characteristic angle is small, but the Lorentzian is broad and the total inelastic scattering is stronger than the elastic. This is further evidence against the PCTF model, but does not detract from the simple aberration correction operation by pixel shifting in tcBF.

HAADF-STEM generated SPA reconstructions of ferritin decorated with isolated zinc or iron atoms (<http://www.pnas.org/content/114/42/11139>). While the protein resolution was poor, about 2 nm, it was sufficient to orient the

particles such that the localization precision of the metal atoms was better than 3 Angstrom. The tcBF approach would probably resolve this discrepancy.

Can the 4D STEM be up-sampled for the E. coli data (2 mrad convergence) while simulating the incoherent or annular bright field modes? Improved contrast would make a convincing case for the tcBF. The case is convincing that EFTEM is not suitable for very thick specimens, so the STEM-equipped investigators have to make a choice among the various STEM modes, rather than between EFTEM and STEM.

The up-sampling is described as a means of improving resolution with a limited number of sampled points in real space, but it could equally be considered as a way to expand the field of view. It would be very useful to present a second table with a list of self-consistent parameters, should someone want to apply the technique in future.

The panel labels for Fig S9 are reversed: a,b refer to ferritin rather than the bacterium as shown. Also, the caption refers to "aberration fitting" in addition to the tcBF shifts. This is not clear. If we consider only defocus, fitting the shifts to the aberration function is just determining the defocus. Is the intent to fitting of higher order aberrations, or is the improvement seen by refinement of the defocus estimate after eliminating the binning in the diffraction space?

Reviewer #2 (Remarks to the Author):

Yu et al. present a new scanning transmission electron microscopy (STEM) approach to study thick specimens. The authors make use of 4D-STEM pixel detectors to work out and demonstrate the tilt-corrected bright field STEM tcBF-STEM technique. The technique makes use of the electron signal detected over entire bright field and corrects their beam tilt by aligning the corresponding individual off-axis bright field images and averaging them. In addition, the authors expedite the approach by reducing the number of required scan steps by a factor of 8 as they interpolate the images. Using tcBF-STEM, elastic as well as inelastically scattered electrons are used for image formation, which is one of the major benefits over conventional cryo-EM approaches that remove the inelastically scattered electrons by an energy filter. The authors show examples and evaluate the contrast of thicker biological specimens of frozen cells. They derive improved contrast and also demonstrate the suitability of single-particle analysis of virus like particles to 7 Å resolution.

The manuscript contains a series of new procedures that are of high interest to the EM-based structural biology. The computation of fast tcBF-STEM images is a significant advance as it is faster than iterative ptychography methods. In fact, tcBF-STEM or parallax have already found their way into applications followed by ptychography. The provided code is well documented and ready to use. The derived PCTF considerations are very insightful for the comparison of STEM methods. However, regarding the accompanied manuscript, the main claims remain anecdotal and the presented data do not demonstrate the benefits. The main quantitative result is that 7 Å with a single-particle averaging can be obtained with the tcBFSTEM approach.

Major:

1. Dose efficiency

The authors claim that tcBF-STEM is more dose-efficient than typical EF-TEM approaches in title and abstract. While this may certainly be a good hypothesis, the authors fail to deliver clear evidence for it. Depending on the definition of dose efficiency, the real benefit will only show when more signal is generated from the same amount of deposited dose. In this case, the authors do not demonstrate that more signal is recorded. This is a clear oversimplification and does not justify the conclusion. As is the dose efficiency is an item suitable for the discussion section of the manuscript but not results. Related in Line 251: How is the fraction of recorded electrons estimated?

2. Sample thickness

As a main point, the authors make claims that within thick samples in title and abstract. They identify more qualitative features in tcBF-STEM rather than EF-TEM. The results from Fig. 1i-l somewhat underwhelming to appreciate. Please plot the claims of signal boost in clear 1D plots showing the bilayer peaks with respect to the background and estimate the effect in a quantitative fashion. The same is true for images presented in Fig. 3. Again with the presented images, a benefit remains anecdotal. The authors estimate the 2-5 fold theoretical advantage in thickness ranges between 400 and 800 nm but fail to demonstrate the estimated effects. As is the sample thickness is an item for the discussion section of the manuscript but not results.

3. Data collection efficiency

A third advantage of the technique is stated as 3-5x improvement in collection efficiency. This conclusion is from the abstract and derived from the use of upsampling. It appears to refer to an improvement in collection efficiency in comparison to EF-TEM. Therefore, it needs to be clearly stated and quantified that the acquisition of tcBF-STEM is still significantly more time consuming than typical EFTEM acquisitions. As is the data collection efficiency is an item for the discussion section of the manuscript.

4. The 7 Å map achieved by tcBF-STEM is the main result of the manuscript. Currently, the only way to judge the map quality is shown in Figure 5. The authors should deposit the maps to the EMDB as it is common in structural biology. This reviewer requests access to the map in order to validate the nominal resolution of the map.

5. Manuscript follows a peculiar hard-to-follow format and requires major re-editing. Main, which reads like a short but rather simplified version of the manuscript, followed by results, discussion. In main, first results are already presented prior to the the introduction. Results and discussions are merged.

In more details: First in main, the result that tcBF-STEM was more capable than EF-TEM is presented without detailing the experimental setup. - line 148: image of mitochondrion? How thick is the specimen? In line 158 estimates are given, how were they determined? For the comparison with EF-TEM, which detector is being used? Is motion correction employed etc? The most critical items need to be reported in the results before a proper comparison can be made and not hidden in the Methods section.

In between there are sections (113-147) that read like theory from a textbook without being an introduction or presenting the data to support the claims.

The section on optical limits and scan resolution attempts to delineate the effects on resolution (195-205). However, there is no clear relationship presented on how to bring the scan step and opening angle together into a common resolution concept. Either omit this part or present a clear concept on the relationship.

6. The authors conclude that the upsampling procedure achieves information transfer up to 7. Where does the factor come from? Is it theoretically derived or based on the gold reflection? A clear derivation of this factor needs to be presented.

7. Code review

a) Installation

code download and installation through the provided anaconda environment file in combination with jupyter notebook is well documented and easy to use. I did not encounter any errors during setup.

b) Code execution

The example notebook provides excellent guidance through the steps of loading in raw data, solving for shifts in the detector geometry, fitting the aberration function and outputting the upsampled tilt-corrected brightfield image. Code performance was acceptable.

c) Example data

The provided raw data examples on <https://zenodo.org/records/10825339> contain three low-dose 4D-STEM EMPAD examples, comparable to the data shown in the manuscript. Repeating the tcBF-STEM calculations was straightforward.

d) Potential improvements:

- Provide example code for storing the upsampled tcBF-STEM image
- Provide example code for batch-processing many images, as this is common in CryoEM
- Provide PAD import routines for other detector data types than the two EMPAD .raw files, eg timepix, medipix, Arina HDF5 (<https://www.dectris.com/en/support/downloads/header-docs/hdf5/>)

The manuscript presents additional aspects of STEM in paragraphs that are surprisingly short and anecdotal. They raise more questions than they address. It is unclear what they add to the manuscript:

8. The inclusion of the iDPC results with the Panther detector are somewhat unusual (Fig. S11). Presented in this manner, they leave the reader with a big question mark. Why did the authors not take greater care in setting up in-focus images? The results are not essential for the manuscript but in the way they are presented one can get the impression that the authors wanted to make the point that tcBF-STEM is superior over the iDPC approach, which the authors were not able to show. The results are not essential to the manuscript and paragraph does not add to the manuscript.

9. The authors attempt to investigate the effect of dose on specimen: no signs of bubbling for STEM after 210 e-/Å² and compare it with literature values from another STEM paper comparing it for CTEM. From this experiment, no conclusions can be drawn as the bubbling rate is very specimen dependent. They also never see bubbling. The results are not essential to the manuscript and paragraph does not add to the manuscript.

Minor:

line 70: "The performance is still worse than EFTEM and TEM when beam-induced motion is corrected, and suggests..."
- take out due to suggestive character from the introduction as it has not been demonstrated that beam-induced motion is the limiting factor.

line 79: why nominal?

- line 138 - 148 redundancies: topic of chromatic blur is introduced again as in line 43

typo: line 366: in summary

Line 562: Given the importance of the CTF including the theoretical derivation, there is no mentioning of the CTF in the cryoSPARC workflow. Can the author add the specific options that were used to address the defocus?

Reviewer #2 (Remarks on code availability):

7. Code review

a) Installation

code download and installation through the provided anaconda environment file in combination with jupyter notebook is well documented and easy to use. I did not encounter any errors during setup.

b) Code execution

The example notebook provides excellent guidance through the steps of loading in raw data, solving for shifts in the detector geometry, fitting the aberration function and outputting the upsampled tilt-corrected brightfield image. Code performance was acceptable.

c) Example data

The provided raw data examples on <https://zenodo.org/records/10825339> contain three low-dose 4D-STEM EMPAD examples, comparable to the data shown in the manuscript. Repeating the tcBF-STEM calculations was straightforward.

d) Potential improvements:

- Provide example code for storing the upsampled tcBF-STEM image
- Provide example code for batch-processing many images, as this is common in CryoEM
- Provide PAD import routines for other detector data types than the two EMPAD .raw files, eg timepix, medipix, Arina HDF5 (<https://www.dectris.com/en/support/downloads/header-docs/hdf5/>)

Reviewer #3 (Remarks to the Author):

The paper describes a new method, tc-BF, to retrieve phase images from 4D STEM data sets and applies and explores the method with a particular focus on biological applications. Phase imaging is very important in electron microscopy of biological materials and in cryo-EM a large defocus is used to generate phase contrast. As the manuscript makes very clear, a specific advantage of retrieving a phase image from 4D STEM is the avoidance of needing to energy filter with elastic contrast being preserved in inelastic scattering, allowing many more transmitted electrons to contribute to the image. Both ptychography and the iDPC method have been demonstrated for retrieving retrieve phase contrast images of biological samples. The beauty of the tc-BF method is that it is actually rather simple and avoids the problems of the iterative reconstruction required for ptychography under conditions of defocus, whereas iDPC cannot be used with defocus so loses efficiency at low spatial frequencies.

The paper is well-written and articulates the advantages of the method well. Of course, the method will only be widely adopted if it can shown that the method can solve a specific biological problem that previous methods cannot. That advance is not shown in the current manuscript. The comparison with the conventional EFTEM approach is quite qualitative – for example the improvement in image quality shown in Figure 1, while just about present, is not particularly striking. A similar comment could be made about the other comparisons presented. The single particle reconstruction presented demonstrates that it can be done, but fails to reach the resolution of the existing EFTEM approach.

I think the method has great potential and I am keen that the work be published. It is an editorial decision whether a specific advance in capability over existing methods needs to be demonstrated to be publishable in Nature Methods, though I note that the previous iDPC demonstration of single particle analysis – reaching 3.5 angstroms - was published in Nature Methods.

More specific comments are:

Line 102: It would be useful to clarify that the word “combined” here is mathematically a summation.

Line 127: Could the authors clarify whether coma is actually corrected in the current work?

Line 260: I am not clear why the elastic signal is said to decrease with thickness with a decay constant given by the elastic mean free path. Could the authors explain why this would be the case. Is it elastic scattering outside the detector area? If so, how was the mean free path calculated?

Line 307: Are the authors saying that tc-BF is unlikely to be competitive for a single particle analysis application. It would be useful to give the reader a clear conclusion here.

Line 376: Low frequency transfer is controlled by both the convergence angle and the defocus. Was there any rational basis to the choice of parameters used here?

Line 508: The DQE plots look sensible, but I am confused by the equations that have the noise power spectrum in the denominator because that would imply that the DQE is dose dependent, which it is not. Is there a normalisation factor here which I am not seeing, perhaps one that sets the DQE to one at zero frequency?

Figure 2: If the 2.3 angstrom Au fringes are in the power spectrum, why is it not possible to see them in the image for this technique?

Figure 2(a): Is the shift measured by cross-correlation? This might be stated somewhere and I might have missed it.

Figure 2(c): Is this image formed from a single detector pixel?

Reviewer #3 (Remarks on code availability):

The code is overall well structured. The example notebook is functional and explains the process. In the "Load data" section the markdown states that the dataset is recorded with a convergence angle of 7mrad and a step size of 2.87 nm. Yet in the following example analysis a convergence angle of 5.5mrad and a step size of 2.88 nm is used. Rerunning the code with the parameters stated in the markdown leads to no errors.

The "EM" directory provides the EM image class which offers basic post processing tools for EM data. However, the example notebook only demonstrates its visualisation functions. All the actual processing is performed using the "PAD" module. It would have been nice to see a few more comments in the "EM" directory.

The PAD directory contains the PAD module, the PADutil module, the display module and the AbrFit module. The core functionality of calculating the shifts is performed using the GitHub directory <https://github.com/bsavitzky/rigidRegistration/> by Ben Savitzky. The AbrFit module provides the aberration fitting functions from 1st order to 3rd order. Separate functions are used depending on the order. It would have been nice to see a unified aberration fitting function where the order is an input parameter. This module was not used in the example notebook, instead the 3rd order fitting functions were rewritten. The directory also contains several other python modules which appear to be unused backups or updates of other modules. The code in the PAD directory is very well documented and could easily be extended.

Conclusively, the key codebase is well documented, functional and well demonstrated using the notebook. Besides further generalisation of the AbrFit module, it would have been nice to either unify the EM and PAD directories removing unused code. Alternatively, the need to keep the code from the two directories separate could have been explained in the README.md file.

*Editor's note: Reviewer #5 was a co-reviewer with Reviewer #3.

Reviewer #5 (Remarks on code availability):

Installation

code download and installation through the provided anaconda environment file in combination with jupyter notebook is well documented and easy to use. We did not encounter any errors during setup

Code execution

The example notebook provides excellent guidance through the steps of loading in raw data, solving for shifts in the detector geometry, fitting the aberration function and outputting the upsampled tilt-corrected brightfield image. Code performance was acceptable.

Example data

The provided raw data examples on <https://zenodo.org/records/10825339> contain three low-dose 4D-STEM EMPAD examples, comparable to the data shown in the manuscript. Repeating the tcBF-STEM calculations was straightforward.

Potential improvements:

- Provide example code for storing the upsampled tcBF-STEM image
- Provide example code for batch-processing many images, as this is common in CryoEM
- Provide PAD import routines for other detector data types than the two EMPAD .raw files, eg timepix, medipix, Arina HDF5 (<https://www.dectris.com/en/support/downloads/header-docs/hdf5/>)
- Minor documentation cleanup eg for the iDPC part

Version 1:

Decision Letter:

Our ref: NMETH-A57547A

17th Mar 2025

Dear Dr. Yu,

Thank you for submitting your revised manuscript "Dose-Efficient Cryo-Electron Microscopy for Thick Samples using Tilt-Corrected Scanning Transmission Electron Microscopy, Demonstrated on Cells and Single Particles" (NMETH-A57547A). It has now been seen by the original referees and their comments are below. The reviewers find that the paper has improved in revision, and therefore we'll be happy in principle to publish it in Nature Methods, pending minor revisions to satisfy the referees' final requests and to comply with our editorial and formatting guidelines.

TRANSPARENT PEER REVIEW

ORCID

Sincerely yours,
Allison

Allison Doerr, Ph.D.
Chief Editor
Nature Methods

Reviewer #1 (Remarks to the Author):

The manuscript is really much improved by the authors response to constructive criticisms from the referees. I'm happy to recommend publication and look forward to seeing it "in print". I have only a few very minor technical comments that I noted while reading.

line 46. energy loss<e> electrons...

line 217-218. cutoff <and> but is just within...

Fig. S7. It appears that the comments starting "The phase ramp across each individual PCTF..." refer specifically to panel (b), which is probably not the intent.

Fig. S9. The numbers in the thickness guide bar are wrapped, all except for 476 at the bottom.

Fig. S14. The labels a and b have the closing parenthesis wrapped.

Fig. S17. TcBF appears with capital T.

Fig. S19. The reference to Kirkland's code is #35.

Reviewer #2 (Remarks to the Author):

The authors have taken the comments very seriously. The manuscript is now consistently formatted. Background is better supported by introductory papers, results and discussions now flow in a logical order. Many of the unsupported claims are now supported by the requested data in a clearer fashion. The side-by-side comparisons are now clearly legible and interpretable, 1D profiles support the observable features for STEM vs EFTEM much more clearly.

I stumbled upon a minor question in Figure S13:

The details on the structure refinement from tcBF are now sufficient due to the helpful map validation report. Figure S13: The presented TEM class average only has low resolution (20 Å) while the FSC goes to very high resolution (3.4 Å). I expect that better class averages are available revealing secondary structures? In that regard, the 2D class average may be misleading and not be representative of the data as the FSC of the TEM reconstruction goes to 3.4 Å resolution.

I would also like to ask the authors to submit the 3.4 Å TEM map to the EMD for future comparisons of TEM and STEM approaches.

Reviewer #2 (Remarks on code availability):

No further comments required.

Reviewer #3 (Remarks to the Author):

I do not think protracted reviewing rounds are helpful, so I will keep this second review relatively short. Overall, I think the authors have addressed the comments I raised, and as I said in my initial review, I am keen to see this work published. There are two aspects of the revised manuscript, however, that I would like clarified before the work can be published.

1. To quantify the improvement in imaging thick samples using tcBF-STEM compared to EFTEM, the authors have introduced some new figures of data in Figure 3. These show the comparison of contrast and signal to noise as a function of sample thickness. The actual methodology of how these measurements were made does not seem to have been explicitly explained and should be added. In particular, looking at the data in Fig. S12, the complex structure in the data means that separating signal from noise is non-trivial and needs to be explained in detail. Also, what is the origin of the dashed lines in the plots in Figure 3. If they do not have a scientific basis then they can be removed.

2. In some of the newly added text, the term signal seems to have been used purely to refer to the number of electrons making it to the image, see for example the sentence starting on line 252 where it states that the signal has dropped to ~14% of the dose. Referring to signal should mean something that carries information such as contrast in an image. You can have plenty of intensity in an image with very little signal. The language around this needs to be tightened up. This comment also applies to Table I, where the fraction of electrons making it to the image is quoted, but this tells us nothing about signal or information. Coherence loss would not reduce the number of electrons but would reduce the signal, for example.

Reviewer #3 (Remarks on code availability):

The code was reviewed in the previous round.

Response to Referees

Manuscript ID: NMETH-A57547

Title: Dose-Efficient Cryo-Electron Microscopy for Thick Samples using Tilt-Corrected Scanning Transmission Electron Microscopy, Demonstrated on Cells and Single Particles

Dear Reviewers and Editor,

We sincerely thank you for the thoughtful and constructive suggestions and comments. With this revision, we have aimed to address all comments relevant to our claims. Specifically, we have restructured the main text, improved the presentation of the figures, better organized the results and discussion sections, and included additional analyses in the supplementary materials.

Below, we provide a point-by-point response to each reviewer's comments. Reviewer comments are quoted in blue, and our responses are provided in black. Unless otherwise specified, all line and figure references in our responses pertain to the revised version.

Reviewer 1 Comments

Main comments

The primary concerns reflect the reviewer's desire to understand the contrast mechanisms in thick samples. Phase contrast works best for thin samples but becomes increasingly diminished as samples get thicker. Currently, there are no robust models of phase contrast for thick samples, yet phase analysis continues to be routinely used in cryo-EM with conventional TEM.

By the theorem of reciprocity, the same theoretical challenges apply to STEM and, by extension, to our method. We acknowledge that we are unable to resolve these challenges in this work. However, we aim to provide the same level of theoretical approximation currently used for TEM, allowing for comparisons to be made on an equal footing. Our key objective is to demonstrate how knowledge and familiarity with TEM can be leveraged to better understand STEM performance. The question of how much we observe as phase versus amplitude contrast is indeed intriguing and warrants further exploration. However, it is not the central focus of this paper and is likely to be examined in greater detail in the future, given the richness of information contained in 4D-STEM datasets.

Overall, the comments about PCTF being too simple for modeling the imaging of thick samples are well taken. This problem is not unique to our method and the electron microscopy community does not have a clear answer that sufficiently captures the complexity of imaging thick samples – we instead provide the same level of approximation as the TEM community uses. We cannot address this unsolved broader

concern in this work, but we have tried to clarify the language to better acknowledge the limits of our theoretical analysis of the method.

My most substantial complaint is that the whole theoretical framework presented in the online methods presumes that the dominant contrast mechanism is actually phase contrast. On the other hand, thick specimens scatter very strongly so amplitude contrast cannot really be ignored (lines 132 and 280 in the manuscript). At the elastic mean free path, which is roughly equal to the thickness of the specimens of interest, the fraction of intensity remaining unscattered is only $1/e$. (Table I is probably “optimistic” in terms of transmission; it is not clear whether the numbers originate from the full field of view or from the dense areas of interest shown in the insets. The table should also display the convergence angle and depth of field for later reference.) Low angle scattering may interfere coherently within the cone of illumination, but beyond that, intensity is lost to the dark field. Indeed, such cellular samples produce strong contrast in annular dark field STEM. Nonetheless, it is clear that the diffraction-space shifting, i.e., tilt correction in the BF, makes a significant improvement in practice. My conclusion is that the protocol works well even beyond the realm in which it is formally derived in the manuscript, and I think this deserves comment. We can speculate that the coherent phase contrast contributes most significantly at the higher spatial frequencies, but as far as I’m aware there is no general solution to this problem.

The request to comment on amplitude contrast is very reasonable. The simplest answer why our approach works even in the presence of significant amplitude contrast is that for a defocused image the amplitude term is blurred out so it only provides a slowly-varying background. (Normally one works in-focus for ADF and IBF, but to get a phase shift we have to defocus.) In figure R1 below, which is 500-600 nm thick across the field of view, the background from the amplitude term is 30-50% of the signal in our samples, while the tcBF signal account for the other 50-70%. As the reviewer notes later, the amplitude term can be determined from scattering outside the aperture – i.e. from a tilt-corrected dark field (tcDF) image. Shown below, the membranes are visible with tcBF and is blurred out in tcDF (and also IBF and ADF), showing the blurring of the amplitude contrast. Poisson noise from this background will degrade contrast, but as it scales as the square root of the background, the effect is not as bad as one would imagine based on a linear scale.

Figure R1. Comparison of different imaging modes for the data set from main figure 3: a) Incoherent Bright field (IBF); b) tilt-correct bright field (tcBF); c) annular dark field (ADF); d) tilt-corrected dark field, all without upsampling, and at 2-micron defocus. At this defocus, the tcBF images retains sharp contrast, and comprise 50-70% of the incident beam. The tcDF formed from electrons scattered outside the aperture is more blurred, as are the IBF and ADF incoherent images.

We have included the discussion above on amplitude contrast included in the manuscript, the discussion section lines 350-356 and Fig.S16 .

(Table I is probably “optimistic” in terms of transmission; it is not clear whether the numbers originate from the full field of view or from the dense areas of interest shown in the insets.

Table I shows the average thickness across the field of view for each region listed in the table as indicated by the column heading. For a more-detailed breakdown of thickness

variations, Suppl. Fig S9 shows a spatially-resolved thickness map. This is summarized in table 1 as an average thickness of 547 nm, and the map itself shows thickness range is ~500-600 nm. Fig R1 above (Fig. S16 in the manuscript) of the same region shows the transmitted fraction ranges from 40-70%. Table I gives an average across the image of 53%. In general the averages across the images are consistently mid-range. This is to be expected when the sample thicknesses are a mean free path or less, as all our samples are. It is only at about 2 mean free paths (i.e. ~1.6 microns thick) that significant non-linearities would dominate.

The table should also display the convergence angle and depth of field for later reference.)

We have added the convergence angle and the iDPC/IBF depth of field to table I. For tcBF, the depth of field (DoF) depends on spatial frequency within the aperture, not the aperture itself.

Probably the simplest STEM method for thick specimens is the incoherent bright field (IBF), and so far it has been the method of choice for cryo-tomography by STEM (<https://doi.org/10.1021/acs.accounts.1c00279>, <https://onlinelibrary.wiley.com/doi/abs/10.1111/mmi.15213>). Integrating the remaining signal on axis is inverse to the dark field, by conservation, but it is less sensitive to multiple elastic scattering (with credit on that point to the Muller group, among others). tcBF should always be “equal to or better than” IBF, and it would be interesting to ask whether or when it actually justifies the substantial effort. For a small convergence angle, the depth of field may be larger than the specimen thickness, for example the bacteria imaged with a semi-convergence of 2 mrad. One might argue, in such a case, that the effect of tilt correction will be marginal, but I suspect it will still be impressive. That would be an interesting comparison because the simplicity of working in focus with a fast single-pixel detector still has clear practical advantage. For a larger convergence angle, with a depth of field shorter than the sample thickness, the incoherent projection will just give a blur and the tilt correction is really unavoidable. The benefit should be an effectively smaller probe with uncompromised lateral resolution.

This work on tcBF is conducted in a very different regime from where IBF would normally be collected. IBF is best in focus where it gives maximal contrast, and tcBF only works out of focus. At zero defocus in a thin sample, there are no shifts in tcBF (or any phase contrast), so the IBF and tcBF signals are equivalent then. At large defocus, the IBF images blur out – for a thin sample it is the “before tilt correction” panel of figure 1f.

For thick samples, we tried a comparison of IBF and tcBF – figure R4 below. The tcBF image is much clearer but we worry it is still not a fair comparison as there is still a small defocus present. We defer this to future work as we suspect the thickness dependence is quite complicated - especially beyond 1 micron where very little phase contrast is left. It will probably require a full paper in itself to explore this regime.

The manuscript goes into great detail on the comparison with iDPC. The text and derivations are very useful, but the emphasis seems a bit excessive.

iDPC has attracted interest from the cryo-EM community (e.g., Lazić et al., Nat Methods, and our Ref 23) as a promising method for phase contrast imaging in STEM, demonstrating results in SPA reconstruction at near-atomic resolution. However, recent work in both iDPC (ref 23) and ptychography (ref 25) shows that both methods fail to transfer low spatial frequencies when using larger apertures. We show that this poor information transfer is the result of a damped DQE, despite a seemingly promising CTF. In contrast, tcBF at large defocus recovers this information, and the derivation suggests ptychography should be attempted at larger defoci to benefit from this information – a point we make towards the end of the discussion.

What I find missing, at least for consistency, is a discussion of the CTF for thick specimens. I interpret Fig 4b as simply the TEM-like CTF at a precise (but unspecified) defocus, presumably 700 nm as simulated in the supplementary figures S5-S7, which should predict the CTF for tcBF. If the sample is half a micron thick, how much does the CTF differ from top to bottom? The zeros would appear at different depths, possibly complicating the interpretation.

We believe this point is indeed worth commenting on and we have added Suppl Fig S14 and S15 showing the CTF vs defocus, and DQE obtained by averaging over the thickness of a 0.5 and 1 micron thick sample vs defocus, with a discussion in the main text on lines 342-349.

The observation that the CTF must change in depth through the sample is correct. Molecules at the top of the sample will have a different CTF from those at the bottom. This also occurs in EFTEM when performing sub-tomogram averaging, and is corrected there by applying a local CTF. The same should be the case here.

However, if we simply average through the projection then the high frequencies are lost, much along the lines predicted by the simple depth of field equation $\Delta z(\alpha) \approx \lambda/\alpha^2$, where α is the spatial frequency of interest. This means measuring the projected amorphous speckle pattern will give a worse estimate of resolution than measurements from individual molecules.

For a 1-micron defocus at the entrance to the sample and a 600 nm thick sample, information in projection is transferred out to about 6-7 Å. (For an individual molecule, the local CTF will have a better information transfer). Again, this theory is very similar to what is used in EFTEM.

700 nm defocus is impractically low for the examples shown, both due to the specimen thickness and the 8-fold up-sampling. The simulation for iDPC, on the other hand, relates to in-focus imaging, so the fair comparison would be to consider a defocus spread over the same thickness.

We are using 700 nm as an example in the simulations, as the larger defocus values have too many zero-crossings to display easily in a small figure, resulting in aliasing in the displayed image when images are compressed. 700 nm was about the largest we could go before displaying information became impractical. The same trends are present for 700 nm or 2000 nm defocus, which is that both show good information transfer at low frequencies, but with many frequent zero crossings.

We have added supplementary figure showing damping for the iDPC CTF and DQE vs a much larger defocus spread (lines 342-349, Fig. S14-15). The damping in iDPC is very much worse, because of its short depth of field. As the reviewer notes below, the very short depth of field of iDPC is a benefit for depth sectioning (the comment that it “shows impressive depth sectioning due to the maximum contrast in focus”).

All that said, it is far from clear to me that the PCTF construction is conceptually a good model for electron imaging through a bacterium. A simple acid-test is that the contrast should invert on passing through focus, which I think is very unlikely. Contrast will be dominated by scattering out of the bright field. Moreover, the angular spread and coherence loss from inelastic scattering are much more severe than usually considered. It is clear, on the other hand, that the geometric shifts of the pixels in the diffraction plane do compensate for optical aberrations.

We only measured through-focal series for thin samples, and they do show contrast reversals as expected, but we unfortunately did not measure a thick sample – it is something we are planning to do in a few months, so perhaps useful for a follow-up paper.

Here is a defocus reversal for Au nanoparticles on a quartz foil SiO₂ supporting film (thickness ~50 nm) showing contrast reversals.

However, there are other signatures of coherent imaging in our existing data sets such as Fresnel fringes around membranes even in thick cell sections (Fig R3 discussed below).

Figure R2. Contrast reversal of Au nanoparticles (red arrows) on a SiO₂ supporting film in tcBF images with -2 μm defocus compared to +2 μm defocus.

The question of whether the PCTF is still useful is essentially a question of whether any coherent image contrast is left, or if the image is wholly incoherent. The condition for coherence is that the collection angle must be smaller than the convergence angle as shown in Fig 3.9 of ref 39:

Fig. 3.9 BF-CTEM transfer function for weak phase objects with increasing condenser angle β_{\max} as a function of dimensionless spatial frequency K (Scherzer defocus and aperture α_{\max}). $\beta_{\max}/\alpha_{\max} = 0.1, 0.2, 0.5, 1.0$. The phase contrast transfer function is zero when the image process is incoherent $\beta_{\max}/\alpha_{\max} \geq 1$

The convergence angles used in this work are $> 4\text{mrad}$, so even for an inelastic spread $\sim 1\text{ mrad}$, the conditions for coherent imaging are comfortably satisfied. We have plotted the degree of partial coherence vs aperture ratio in Suppl. Fig. 1. For a 4 mrad convergence, and a 1 mrad collection blur, we would expect a 20% reduction in peak contrast. This is discussed on lines 173-181.

As further evidence of coherent imaging in our existing datasets (Fig R3), we observe features such as Fresnel fringes around membranes, even in thick cell sections. For example, the inset image of ribosomes in 600-800 nm thick *E. coli* cells shows Fresnel fringes (indicated by red arrows pointing to the white fringes). Additionally, Fresnel fringes around the mitochondria are visible in Figure 3 of the manuscript. Without phase contrast, such features would not be observable.

Figure R3. Fresnel fringes around ribosomes in 600-800 nm thick *E. coli* cells (inset), indicated by red arrows pointing to the white fringes, and Fresnel fringes around the mitochondrion, as shown in Figure 1 of the manuscript.

In practice, thick samples are difficult to interpret not only because of the optical imaging conditions, but also because distinct specimen features “pile up” in a projection. One way to address this is with a very large convergence angle, even in ADF-STEM (<https://onlinelibrary.wiley.com/doi/epdf/10.1002/sml.202402871>). Another is by tomography, using a smaller convergence to get a nearly parallel projection. A third is actually iDPC (<https://www.sciencedirect.com/science/article/pii/S1047847722000077>), which shows impressive depth sectioning due to the maximum contrast in focus. Parallax deshifting tends to extend the depth of field (<https://www.cambridge.org/core/journals/microscopy-and-microanalysis/article/flexible-stem-with-simultaneous-phase-and-depth-contrast/131F0EAA69FD8D9665E7E685EC81B1BE#>), at least if we can extrapolate from quadrant detector data.

Yes, this is covered in references 14-18 (the parallax paper is ref 18). tcBF is intended for the tomography approach.

Technical comments

The life science readership will be confused by some of the notation. This begins in Fig 1, where the microscope component labels follow the dedicated STEM nomenclature. The objective aperture, for example, is the condenser C2 aperture for TEM. Since life science is presumably a target audience, this should just be made clear.

We have updated Fig.1b, we re-labelled the probe-forming aperture in STEM mode.

Inelastic scattering is often considered as described in the text: strongly forward-peaked and basically a perturbation on the elastic-scattering STEM signal. For the light elements

of organic materials this is a problematic assumption. (<https://www.sciencedirect.com/science/article/pii/S0304399124000159>) The characteristic angle is small, but the Lorentzian is broad and the total inelastic scattering is stronger than the elastic. This is further evidence against the PCTF model, but does not detract from the simple aberration correction operation by pixel shifting in tcBF.

We appreciate the comment because the larger median angle does set a limit as to what the smallest useful pixel on the detector would be (~0.7-1 mrad). It also serves as a reminder that the convergence angle should be kept above ~ 4mrad to preserve coherence.

The recent paper cited above by Elbaum and coworkers measures about 8% of the inelastic scattering within the characteristic angle, and measures a median inelastic angle of ~1.4 mrad at 120 or 200 keV. This is consistent with the EELS literature (see ref 8). Both these angles are much smaller than the characteristic elastic scattering angle which is ~10 mrad. In other words, inelastic scattering is more strongly peaked in the forward direction than elastic scattering or to quote directly from ref 8, pg 126 “inelastic scattering is therefore concentrated into considerably smaller angles than elastic scattering, as seen from Figs. 3.5 and 3.7.” These are the conditions needed to preserve some coherent contrast.

Ref 8, sections 3.3.1.5 and 5.5.3 also gives an analytic model for the median inelastic angle, which is 0.7 mrad at 300 keV (the angle decreases with increasing beam energy). Ref 13 finds a similar angular spread at 300 keV.

For coherent imaging, the angular spread on the detector must be smaller than the convergence angle of the probe (see Fig 3.9 of reference 39), which is 4-6 mrad, thus this condition is still met, even after multiple inelastic events, so some coherent contrast remains.

We discuss this on 173-181, and include the reviewer's reminder that the median angle is larger than the most probable angle, but that both are still smaller than the convergence angles we use.

HAADF-STEM generated SPA reconstructions of ferritin decorated with isolated zinc or iron atoms (<http://www.pnas.org/content/114/42/11139>). While the protein resolution was poor, about 2 nm, it was sufficient to orient the particles such that the localization precision of the metal atoms was better than 3 Angstrom. The tcBF approach would probably resolve this discrepancy.

We agreed that this would make an important follow-up paper.

Can the 4D STEM be up-sampled for the E. coli data (2 mrad convergence) while simulating the incoherent or annular bright field modes? Improved contrast would make

a convincing case for the tcBF. The case is convincing that EFTEM is not suitable for very thick specimens, so the STEM-equipped investigators have to make a choice among the various STEM modes, rather than between EFTEM and STEM.

Here, we compare IBF and ADF with tcBF for the *E. coli* dataset using a 2 mrad convergence angle. Panel (a) illustrates the angular ranges used for ADF (green annulus) and for IBF and tcBF (red circle). The combination of initial defocus and sample thickness exceeds the depth of field for both ADF and IBF. In contrast, tcBF retains clear contrast within the cell. Ideally this should be redone with the IBF in focus to remove any doubt.

Figure R4. Panel (a) shows the angular ranges for ADF (green annulus) and IBF/tcBF (red circle). The corresponding images are shown for ADF in (b), IBF in (c), and tcBF in (d).

The up-sampling is described as a means of improving resolution with a limited number of sampled points in real space, but it could equally be considered as a way to expand the field of view. It would be very useful to present a second table with a list of self-consistent parameters, should someone want to apply the technique in future.

We added a Fig. S2 and lines 402-414 in discussion explaining the link between upsampling, defocus Δf and convergence angle α . In general the step size Δd should be kept smaller than $\Delta d = \alpha \Delta f$, and the upsampled pixel size will be $\Delta \alpha \Delta f$ where $\Delta \alpha$ is the angular pixel size on the STEM diffraction pattern (or the inelastic median angle if larger). This means the degree of upsampling possible is $\alpha / \Delta \alpha$.

The panel labels for Fig S9 are reversed: a,b refer to ferritin rather than the bacterium as shown.

Thank you for catching this. It was our mistake.

Also, the caption refers to “aberration fitting” in addition to the tcBF shifts. This is not clear. If we consider only defocus, fitting the shifts to the aberration function is just determining the defocus. Is the intent to fitting of higher order aberrations, or is the improvement seen by refinement of the defocus estimate after eliminating the binning in the diffraction space?

The algorithm does measure higher-order aberrations, but their impact depends on the maximum angle in the probe. For a 2-4 mrad probe, usually only defocus, astigmatism and coma matter. For a well-aligned instrument at a large defocus, defocus should dominate. At 30 mrad in a corrected instrument, aberrations up to C5 or C7 can be measured.

Reviewer 2 Comments

Main comments

1. Dose efficiency

The authors claim that tcBF-STEM is more dose-efficient than typical EF-TEM approaches in title and abstract. While this may certainly be a good hypothesis, the authors fail to deliver clear evidence for it. Depending on the definition of dose efficiency, the real benefit will only show when more signal is generated from the same amount of deposited dose. In this case, the authors do not demonstrate that more signal is recorded. This is a clear oversimplification and does not justify the conclusion. As is the dose efficiency is an item suitable for the discussion section of the manuscript but not results.

This was our mistake in making the key figure a small inset that lost a lot of detail in the compressed PDF for review. We have done redone the figures to make this larger and clearer (Fig 3a-d of text, Fig R5 below). We have also added quantitative measures.

For the dose efficiency comparison, we ensured the same incident dose over vacuum for both EFTEM and tcBF, and compared the collected signal (lines 251-252, table I) – “Under the same incident dose, with direct electron detectors used for both techniques, the collected electron counts were compared for dose efficiency.”

We have added Fig. 3e,f (Figure R6 below), Fig.S11-S12, lines 256-260 to show the contrast and signal/noise ratios for fringe contrast on the double membranes as a function of sample thickness. The raw line profiles are also submitted along with this revision in .csv format. We hope these new figures speak for themselves.

Related in Line 251: How is the fraction of recorded electrons estimated?

For both techniques, direct electron detectors were used and the signals collected are calibrated in electron counts for comparison. (lines 251-252, table I).

2. Sample thickness

As a main point, the authors make claims that within thick samples in title and abstract. They identify more qualitative features in tcBF-STEM rather than EF-TEM. The results from Fig. 1i-l somewhat underwhelming to appreciate.

This is our mistake! —the figures lost significant detail after compression in the PDF. To address this, we have made larger figures Fig.3, Fig.S10 and Fig.S11 . Figure R5 provides an enlarged version of the same comparison shown Fig.3, offering additional detail.

Figure R5. tcBF-STEM (red) and EFTEM (green) images of a mitochondrion, presented as zoomed-in versions of the same images shown in manuscript Fig. 3, now with enhanced detail.

Please plot the claims of signal boost in clear 1D plots showing the bilayer peaks with respect to the background and estimate the effect in a quantitative fashion.

We have included line profiles across bilayers in different thickness regions in Fig.S11 to Fig.S12, and comparisons in Fig.3e-f, lines 256-260 in the result section (Fig R6). Raw lineprofile data and the code to analyze them are included in the submission of this revision.

Figure R6 From the fit to the line profiles, we extracted the fringe contrast with respect to the background, and the fringe signal-to-(background noise), which are summarized here in panels (e) and (f) respectively, again showing better performance for tcBF over EFTEM with increasing sample thickness.

The same is true for images presented in Fig. 3. Again with the presented images, a benefit remains anecdotal. The authors estimate the 2 -5 fold theoretical advantage in thickness ranges between 400 and 800 nm but fail to demonstrate the estimated effects. As is the sample thickness is an item for the discussion section of the manuscript but not results.

The new figure 3 gives quantitative measures. These particular figures were intended as a large anecdotal collection because quantitative resolution metrics are just not meant to be applied in such thick samples. These figures are now moved to Suppl Fig. S10. Panels c,d for instance shows what to us seems like very clear differences of features that are detected in STEM and not in EFTEM (with little arrows to help). We hope this would help to see these differences.

3. Data collection efficiency

A third advantage of the technique is stated as 3-5x improvement in collection efficiency. This conclusion is from the abstract and derived from the use of upsampling. It appears to refer to an improvement in collection efficiency in comparison to EF-TEM. Therefore, it needs to be clearly stated and quantified that the acquisition of tcBF-STEM is still significantly more time consuming than typical EFTEM acquisitions. As is the data collection efficiency is an item for the discussion section of the manuscript.

The improvement in collection efficiency refers to the improved collected dose efficiency because 3-5x more of the incident dose is collected as shown in Fig 5a. This does not depend on the upsampling.

We added a discussion on the speed differences in lines 389-394. At present, tcBF-STEM is slower than EFTEM, but this was a function of the detector used. Already STEM detectors that at run at 100 kHz are available. These run 10x faster than the detector used in the paper, and would make the speeds comparable. Within a year,

commercial event-driven detectors will increase the speed yet another factor of 10, at which point STEM would be faster. However, for both TEM and STEM detectors, the limiting factor is the bandwidth off chip, which would imply comparable information rates for both TEM and STEM if using the same generation of detector technology.

4. The 7 Å map achieved by tcBF-STEM is the main result of the manuscript. Currently, the only way to judge the map quality is shown in Figure 5. The authors should deposit the maps to the EMDB as it is common in structural biology. This reviewer requests access to the map in order to validate the nominal resolution of the map.

The primary map, half maps, FSC and the mask used to generate the FSC have now been deposited to EMDB-48236 .

5. Manuscript follows a peculiar hard-to-follow format and requires major re-editing. Main, which reads like a short but rather simplified version of the manuscript, followed by results, discussion. In main, first results are already presented prior to the the introduction. Results and discussions are merged.

We reorganized the manuscript so there is a background section to lay out the theoretical background with additional references to original papers and standard text books, and we moved the first result from Fig 1 to Fig 3 after background is explained. We have also moved part of discussion from the result section to the discussion section. The estimated thickness is included in table 1.

Detailed comments

In more details: First in main, the result that tcBF-STEM was more capable than EF-TEM is presented without detailing the experimental setup.

We have the experimental parameters now summarized in table 1, with details in the method section.

- line 148: image of mitochondrion? How thick is the specimen? In line 158 estimates are given, how were they determined?

Now in lines 234-236: "A full thickness map of this region can be found in Fig. S9 and the thickness is estimated with the inelastic MFP using the EFTEM dataset."

For the comparison with EF-TEM, which detector is being used?

The detectors used in Fig 3 and Suppl Fig S10 are listed in table I and the online method section. EMPAD (not EMPAD-G2) was used for tcBF and a Falcon 4i detector with a Selectris X energy filter was used for EFTEM.

Is motion correction employed etc?

Motion correction was employed for all of the EFTEM analysis, both the comparative analysis of the thick specimens (lines 450-452) and the single particle analysis. All the tcBF analysis does not include motion correction.

The most critical items need to be reported in the results before a proper comparison can be made and not hidden in the Methods section.

In between there are sections (113-147) that read like theory from a textbook without being an introduction or presenting the data to support the claims.

We have expanded the background theory section and added more references

The section on optical limits and scan resolution attempts to delineate the effects on resolution (195-205). However, there is no clear relationship presented on how to bring the scan step and opening angle together into a common resolution concept. Either omit this part or present a clear concept on the relationship.

We have added a derivation of how opening angle, scan step size and defocus are coupled to the methods section (lines 402-414), as well as Suppl. Fig 2.

6. The authors conclude that the upsampling procedure achieves information transfer up to 7. Where does the factor come from? Is it theoretically derived or based on the gold reflection? A clear derivation of this factor needs to be presented.

Upsampling transfer: this is defined by the furthest information transfer (2.3 Å) comparing to the real-space Nyquist sampling limit (lines 209-212). Real space Nyquist limit is set by the scan step size, so the factor of 7 is from the ratio of the scan step size to the optical information limit ($16\text{Å}/2.3\text{Å} \approx 6.957$). The Gold spacing happens to be very close to the optical limit, so it is a useful measure and demonstration of our measured information limit. A second limit is the ratio of aperture size α to angular blur on the detector, $\Delta\alpha$, gives maximum upsampling ratio of $\alpha/\Delta\alpha$. This is discussed on lines 402-414.

7. Code review

a) Installation

code download and installation through the provided anaconda environment file in combination with jupyter notebook is well documented and easy to use. I did not encounter any errors during setup.

b) Code execution

The example notebook provides excellent guidance through the steps of loading in raw data, solving for shifts in the detector geometry, fitting the aberration function and outputting the upsampled tilt-corrected brightfield image. Code performance was acceptable.

c) Example data

The provided raw data examples on <https://zenodo.org/records/10825339> contain three low-dose 4D-STEM EMPAD examples, comparable to the data shown in the manuscript. Repeating the tcBF-STEM calculations was straightforward.

d) Potential improvements:

- Provide example code for storing the upsampled tcBF-STEM image*
- Provide example code for batch-processing many images, as this is common in CryoEM*
- Provide PAD import routines for other detector data types than the two EMPAD .raw files, eg timepix, medipix, Arina HDF5*
(<https://www.dectris.com/en/support/downloads/header-docs/hdf5/>)

The manuscript presents additional aspects of STEM in paragraphs that are surprisingly short and anecdotal. They raise more questions than they address. It is unclear what they add to the manuscript:

8. The inclusion of the iDPC results with the Panther detector are somewhat unusual (Fig. S11). Presented in this manner, they leave the reader with a big question mark. Why did the authors not take greater care in setting up in-focus images? The results are not essential for the manuscript but in the way they are presented one can get the impression that the authors wanted to make the point that tctcBF-STEM is superior over the iDPC approach, which the authors were not able to show. The results are not essential to the manuscript and paragraph does not add to the manuscript.

We have removed this comparison from the manuscript. We do note that we were unable to acquire at focus for the iDPC data due to variations in the sample height. tcBF collection on the same samples was forgiving of defocus errors (and self-calibrating). This issue arose from the data acquisition operator's inexperience with optimizing iDPC data acquisition for biological samples.

9. The authors attempt to investigate the effect of dose on specimen: no signs of bubbling for STEM after 210 e-/Å² and compare it with literature values from another STEM paper comparing it for CTEM. From this experiment, no conclusions can be drawn as the bubbling rate is very specimen dependent. They also never see bubbling. The results are not essential to the manuscript and paragraph does not add to the manuscript.

We wanted to qualitatively check the damage versus dose behavior for tcBF. This has now been moved to the online methods. The observation of bubbling is a brief

anecdotal aside to the damage study, but it has also been noted in previous studies using modern STEM methods (Elbaum, Ref 16). We included this mention to add another data point and, hopefully, inspire a more systematic STEM study of damage mechanisms, which remains far less developed compared to EFTEM studies.

Reviewer 3 comments

Main comments

Reviewer #3 comments that we have not solved a particular outstanding problem in biology using this method and we accept that. We would highlight that this is a methods paper, and we only aim to describe our imaging method and assess its performance and compare it to the state of the art in the field. We believe that this method has the potential to overcome longstanding challenges in biological imaging with its improved dose efficiency, better contrast from thick samples, and improved information limit compared to TEM on uncorrected microscopes. Several other groups are currently using this approach, or variations of it, in biological research, but as yet there is no literature describing the technical aspects of the method outside of our earlier conference abstracts.

Technical comments

Line 102: It would be useful to clarify that the word “combined” here is mathematically a summation.

Yes, they are summed. We have now clarified this in line 136 hopefully to avoid confusion.

Line 260: I am not clear why the elastic signal is said to decrease with thickness with a decay constant given by the elastic mean free path. Could the authors explain why this would be the case. Is it elastic scattering outside the detector area? If so, how was the mean free path calculated?

Yes, the elastic scattering is largely responsible for scattering outside the aperture. The mean free path is defined as $\lambda = 1/n\sigma$ where n is the number density of scatterers and σ is the cross-section of interest, in this case elastic scattering outside the aperture. It was calculated by a multislice calculation on thick model of amorphous ice described in lines 314-316 to be 774 ± 45 nm, and we measure it to be 830 ± 50 nm. Plots as a function of scattering angle are now given in Suppl. Fig. S19. If inelastic scattering contributed significantly, the measured value would have been smaller than the elastic-only simulation.

Line 307: Are the authors saying that tc-BF is unlikely to be competitive for a single particle analysis application. It would be useful to give the reader a clear conclusion here.

While this manuscript shows that tcBF SPA is not yet comparable to EFTEM SPA, it does not conclude that 4D-STEM SPA will be uncompetitive in the future. We have now summarized in (lines 367-374) what could potentially overcome the current limitations, such as improving specimen motion correction in STEM mode.

Line 376: Low frequency transfer is controlled by both the convergence angle and the defocus. Was there any rational basis to the choice of parameters used here?

We added a detailed derivation as to the coupling and choice of parameters such as defocus and convergence angle in lines 402-414, and Fig.S2. The updated table 1 also reports convergence angles, probe sizes used.

Line 508: The DQE plots look sensible, but I am confused by the equations that have the noise power spectrum in the denominator because that would imply that the DQE is dose dependent, which it is not. Is there a normalisation factor here which I am not seeing, perhaps one that sets the DQE to one at zero frequency?

The DQE can be (and usually is) less than 1 at zero frequency. In the definition of the frequency-dependent DQE equation A14, by convention (e.g. ref 49) the CTF and NPS are normalized to peak at 1, and other losses are absorbed into DQE(0). We have now stated that explicitly on line 614.

Figure 2: If the 2.3 angstrom Au fringes are in the power spectrum, why is it not possible to see them in the image for this technique?

The vertical axis of the power spectrum is on a log scale. The Au fringes, being very close to the information limit, are many orders of magnitude too weak to be well resolved against the real-space broadband noise background, but are detectable after signal averaging in Fourier space.

Figure 2(a): Is the shift measured by cross-correlation? This might be stated somewhere and I might have missed it.

Yes. Best illustrated in the code and also mentioned in Fig1 caption -- "For every detector pixel, the image shifts determined through cross-correlation with the on-axis detector pixel are shown by the arrows overlaid on the averaged CBED pattern in (e) with a zoom-in and binned view in (g)."

Figure 2(c): Is this image formed from a single detector pixel?

Fig.2(c) is not formed from a single detector pixel. As the caption indicates, it is a tcBF-STEM image before upsampling. To form a tcBF-STEM imaging, nearly all bright field signal (except the signal near the aperture edge) is used instead of a single detector pixel.

Reviewer #3 (Remarks on code availability):

The code is overall well structured. The example notebook is functional and explains the process. In the "Load data" section the markdown states that the dataset is recorded with a convergence angle of 7mrad and a step size of 2.87 nm. Yet in the following

example analysis a convergence angle of 5.5mrad and a step size of 2.88 nm is used. Rerunning the code with the parameters stated in the markdown leads to no errors.

We apologize for the inconsistency. This is a mistake. The convergence angle should be 7mrad and the step size should be 2.87 nm.

The “EM” directory provides the EM image class which offers basic post processing tools for EM data. However, the example notebook only demonstrates its visualisation functions. All the actual processing is performed using the “PAD” module. It would have been nice to see a few more comments in the “EM” directory.

The EM package was developed as a general supportive package, not only for tcBF. For concience the authors used the visualization from that package. For more details on the EM package, please reach out to the authors.

The PAD directory contains the PAD module, the PADutil module, the display module and the AbrFit module. The core functionality of calculating the shifts is performed using the GitHub directory <https://github.com/bsavitzky/rigidRegistration/> by Ben Savitzky. The AbrFit module provides the aberration fitting functions from 1st order to 3rd order. Separate functions are used depending on the order. It would have been nice to see a unified aberration fitting function where the order is an input parameter.

Examples using the aberration fitting functions aren't included in the example notebook but the authors can provide some upon request.

This module was not used in the example notebook, instead the 3rd order fitting functions were rewritten. The directory also contains several other python modules which appear to be unused backups or updates of other modules. The code in the PAD directory is very well documented and could easily be extended.

Conclusively, the key codebase is well documented, functional and well demonstrated using the notebook. Besides further generalisation of the AbrFit module, it would have been nice to either unify the EM and PAD directories removing unused code. Alternatively, the need to keep the code from the two directories separate could have been explained in the README.md file.

**Editor's note: Reviewer #5 was a co-reviewer with Reviewer #3.*

Reviewer #5 (Remarks on code availability):

Installation

code download and installation through the provided anaconda enviroment file in combination with jupyter notebook is well documented and easy to use. We did not encounter any errors during setup

Code execution

The example notebook provides excellent guidance through the steps of loading in raw data, solving for shifts in the detector geometry, fitting the aberration function and outputting the upsampled tilt-corrected brightfield image. Code performance was acceptable.

Example data

The provided raw data examples on <https://zenodo.org/records/10825339> contain three low-dose 4D-STEM EMPAD examples, comparable to the data shown in the manuscript. Repeating the tcBF-STEM calculations was straightforward.

Potential improvements:

- Provide example code for storing the upsampled tcBF-STEM image

We can provide example code for storing the image upon request. We also note that the final image is produced as a numpy array and can be easily saved using the common matplotlib or tiff file packages.

- Provide example code for batch-processing many images, as this is common in CryoEM

We agree this would be very helpful for the cryoEM processing. We will provide examples upon request.

- Provide PAD import routines for other detector data types than the two EMPAD .raw files, eg timepix, medipix, Arina HDF5 (<https://www.dectris.com/en/support/downloads/header-docs/hdf5/>)

Currently the tcBF package only supports EMPAD and EMPAD-G2. Will consider support more camera's file formats for the future. Thanks for suggesting.

- Minor documentation cleanup eg for the iDPC part

While we wanted to focus on delivering our code for tcBF part, we can provide details / guidance for the iDPC part upon request.

Response to Referees

Manuscript ID: NMETH-A57547A

Dear Reviewers and Editor,

Below, we provide a point-by-point response to each reviewer's comments. Reviewer comments are quoted in blue, and our responses are provided in black.

Reviewer #1:

Remarks to the Author:

The manuscript is really much improved by the authors response to constructive criticisms from the referees. I'm happy to recommend publication and look forward to seeing it "in print". I have only a few very minor technical comments that I noted while reading.

line 46. energy loss<e> electrons...

Corrected in the revision.

line 217-218. cutoff but is just within...

Corrected in the revision.

Fig. S7. It appears that the comments starting "The phase ramp across each individual PCTF..." refer specifically to panel (b), which is probably not the intent.

We updated the caption to avoid confusion.

Fig. S9. The numbers in the thickness guide bar are wrapped, all except for 476 at the bottom.

Corrected in the revision.

Fig. S14. The labels a and b have the closing parenthesis wrapped.

Corrected in the revision.

Fig. S17. TcBF appears with capital T.

Corrected in the revision.

Fig. S19. The reference to Kirkland's code is #35.

Corrected in the revision.

Reviewer #2:

Remarks to the Author:

The authors have taken the comments very seriously. The manuscript is now consistently formatted. Background is better supported by introductory papers, results and discussions now flow in a logical order. Many of the unsupported claims are now supported by the requested data in a clearer fashion. The side-by-side comparisons are now clearly legible and interpretable, 1D profiles support the observable features for

STEM vs EFTEM much more clearly.

I stumbled upon a minor question in Figure S13:

The details on the structure refinement from tcBF are now sufficient due to the helpful map validation report. Figure S13: The presented TEM class average only has low resolution (20 Å) while the FSC goes to very high resolution (3.4 Å). I expect that better class averages are available revealing secondary structures? In that regard, the 2D class average may be misleading and not be representative of the data as the FSC of the TEM reconstruction goes to 3.4 Å resolution.

This is a very fair point. The 2D class average shown is an intermediate step in the processing and doesn't reflect the 3D map's resolution. A 3D map is included now in the figure.

I would also like to ask the authors to submit the 3.4 Å TEM map to the EMD for future comparisons of TEM and STEM approaches.

Submitted to EMDB- 70328.

Remarks on code availability:

No further comments required.

Reviewer #3:

Remarks to the Author:

I do not think protracted reviewing rounds are helpful, so I will keep this second review relatively short. Overall, I think the authors have addressed the comments I raised, and as I said in my initial review, I am keen to see this work published. There are two aspects of the revised manuscript, however, that I would like clarified before the work can be published.

1. To quantify the improvement in imaging thick samples using tcBF-STEM compared to EFTEM, the authors have introduced some new figures of data in Figure 3. These show the comparison of contrast and signal to noise as a function of sample thickness. The actual methodology of how these measurements were made does not seem to have been explicitly explained and should be added. In particular, looking at the data in Fig. S12, the complex structure in the data means that separating signal from noise is non-trivial and needs to be explained in detail.

We update in the FigS12 caption, that the details are in the code, which is provided in the manuscript submission.

Also, what is the origin of the dashed lines in the plots in Figure 3. If they do not have a scientific basis then they can be removed.

The dashed lines are exponential fits that are intended as a guide to the eye. We added a sentence in the figure caption and kept the lines.

2. In some of the newly added text, the term signal seems to have been used purely to refer to the number of electrons making it to the image, see for example the sentence

starting on line 252 where it states that the signal has dropped to ~14% of the dose. Referring to signal should mean something that carries information such as contrast in an image. You can have plenty of intensity in an image with very little signal. The language around this needs to be tightened up. This comment also applies to Table I, where the fraction of electrons making it to the image is quoted, but this tells us nothing about signal or information. Coherence loss would not reduce the number of electrons but would reduce the signal, for example.

We changed the word “signal” to “recorded electrons in the EFTEM images” in line 269 and similar for the word “signal” to “electrons” in line 271.

*Remarks on code availability:
The code was reviewed in the previous round.*

*Reviewer #5:
None*